# PCPO: Proportionate Credit Policy Optimization for Aligning Image Generation Models

**Jeongjae Lee & Jong Chul Ye**
KAIST
{jaylee2000,jong.ye}@kaist.ac.kr

## Abstract

While reinforcement learning has advanced the alignment of text-to-image (T2I) models, state-of-the-art policy gradient methods are still hampered by training instability and high variance, hindering convergence speed and compromising image quality. Our analysis identifies a key cause of this instability: disproportionate credit assignment, in which the mathematical structure of the generative sampler produces volatile and non-proportional feedback across timesteps. To address this, we introduce *Proportionate Credit Policy Optimization* (PCPO), a framework that enforces proportional credit assignment through a stable objective reformulation and a principled reweighting of timesteps. This correction stabilizes the training process, leading to significantly accelerated convergence and superior image quality. The improvement in quality is a direct result of mitigating model collapse, a common failure mode in recursive training. PCPO substantially outperforms existing policy gradient baselines on all fronts, including the state-of-the-art DanceGRPO. Code is available at `https://github.com/jaylee2000/pcpo/`.

## 1 Introduction

Modern T2I generation, dominated by powerful diffusion and flow models (Esser et al., 2024; Labs et al., 2025; Podell et al., 2023), still struggles to create outputs that consistently align with human preferences (Google, 2024). Group Relative Policy Optimization (GRPO), a form of Reinforcement Learning from Human Feedback (RLHF) highly successful in large language models (LLMs) (DeepSeek-AI et al., 2025; Shao et al., 2024), has emerged as the state-of-the-art online policy gradient framework for aligning image generation models (He et al., 2025; Liu et al., 2025; Xue et al., 2025). Despite their success, GRPO methods often encounter training instability and model collapse, limiting their performance and reliability.

In this work, we found that these issues stem from two fundamental limitations that arise when applying policy gradients to generative samplers. First, the standard objective is susceptible to numerical precision errors that skew gradient magnitudes. Second, and more critically, the mathematical structure of these samplers leads to *disproportionate credit assignment*. This manifests as a high-variance learning signal with volatile, non-proportional feedback across timesteps—a primary source of instability that is highly detrimental to the training process.

To address this, we introduce *Proportionate Credit Policy Optimization* (PCPO), a framework that targets both limitations. PCPO first enhances numerical stability by reformulating the objective and, more importantly, ensures proportional credit assignment with a principled reweighting schedule. These targeted modifications accelerate convergence and produce superior samples by mitigating model collapse.

Our work is concurrent to several others aiming to improve alignment efficacy by addressing suboptimal credit assignment. For instance, TempFlow-GRPO (He et al., 2025) uses trajectory branching and MixGRPO (Li et al., 2025) employs a sliding SDE window to focus optimization on high-impact timesteps, primarily for training acceleration. While TempFlow-GRPO also proposes a proportional reweighting scheme, our proportionality principle (eg. Proposition 2) offers a more fundamental explanation for these improvements, successfully accounting for cases where simpler, empirical heuristics fail. As such, PCPO stabilizes the training process, leading to significantly accelerated convergence and superior image quality, and mitigating mode collapse. Experimental results confirm

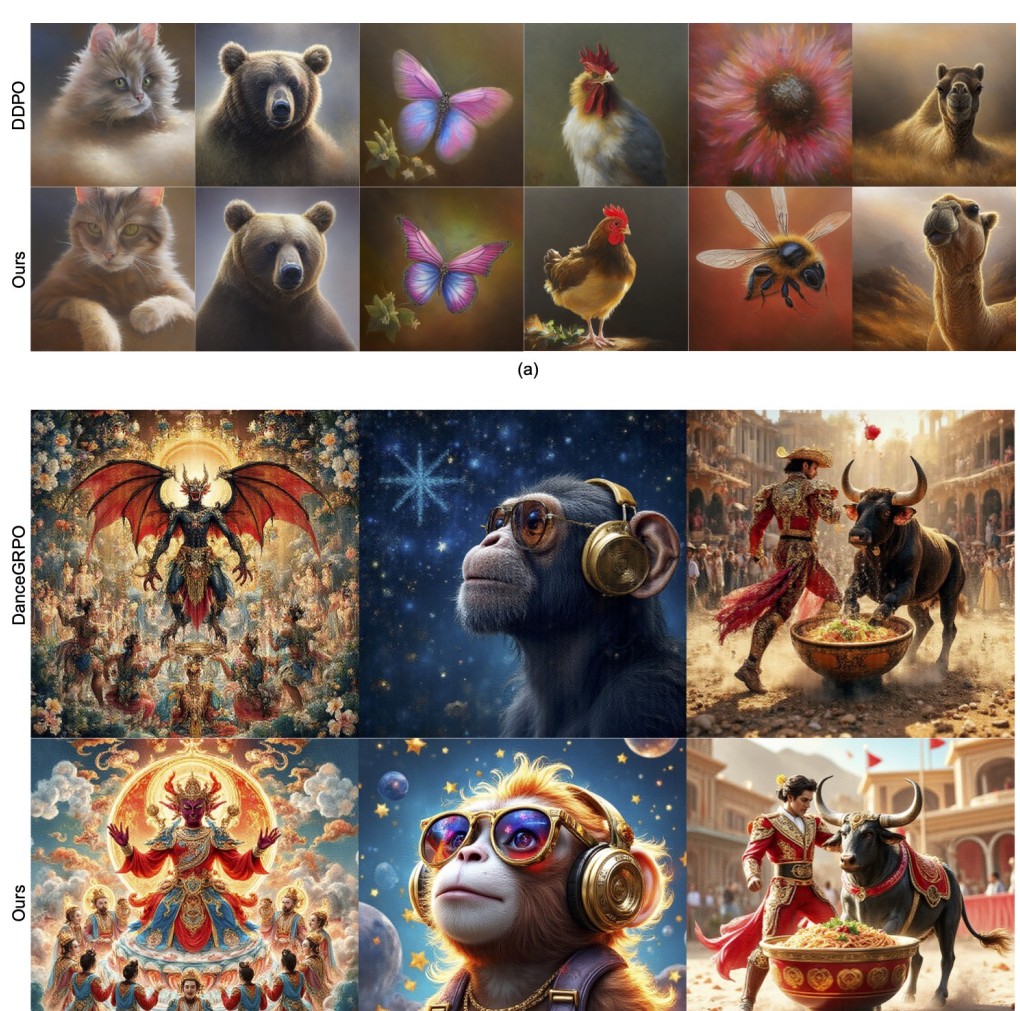

Figure 1: Qualitative comparison of baseline methods (top) and PCPO (bottom) on identical prompts and seeds. PCPO mitigates model collapse seen in baselines across different frameworks. **(a) DDPO (SD1.5, Aesthetics):** At a matched reward level, PCPO preserves diversity and fidelity while DDPO collapses into a blurry, homogenous style. **(b) DanceGRPO (FLUX, HPSv2.1):** After training for 200 epochs, PCPO achieves both a higher reward and superior image quality, avoiding artifacts observed in the baseline.

that PCPO substantially outperforms existing policy gradient baselines on all fronts, including the state-of-the-art DanceGRPO.

**Related Work.** Aligning LLMs with human preferences is predominantly achieved through RLHF (Christiano et al., 2017). Early methods popularized Proximal Policy Optimization (PPO) for this purpose (Ouyang et al., 2022; Schulman et al., 2017). Subsequently, Direct Preference Optimization (DPO) emerged as a simpler, reward-free alternative that gained widespread adoption by reframing alignment as a supervised learning problem on pairwise preferences (Rafailov et al., 2023). However, recent advancements have demonstrated the superior performance of policy-gradient methods: notably, GRPO has surpassed previous techniques on complex reasoning tasks, establishing a new state-of-the-art (DeepSeek-AI et al., 2025; Shao et al., 2024).

The evolution of preference alignment in T2I models has mirrored the trends in LLMs. Initial efforts adapted PPO to the diffusion process but were often plagued by training instability, limiting their scope to constrained vocabularies (Black et al., 2024; Fan et al., 2023). The subsequent adaptation

of DPO improved stability and broadened vocabulary coverage (Wallace et al., 2024; Yang et al., 2024). Most recently, GRPO-based frameworks have achieved state-of-the-art performance (Liu et al., 2025; Xue et al., 2025). This progression was theoretically anticipated; as the optimal policy of DPO is upper-bounded by that of policy-gradient methods (Xu et al., 2024), stabilizing policy-gradient training was expected to yield superior results.

A primary obstacle to enhancing stability and performance of policy-gradient training is model collapse (Shumailov et al., 2024), a degenerative process where a model trained recursively on its own outputs progressively degrades. In the context of online RL for T2I models, we observe this phenomenon manifesting in two key failure modes. The first is classic *mode collapse*, a loss of sample diversity that is also well-documented in LLM alignment, where the policy's entropy is exhausted in pursuit of high rewards (Cui et al., 2025; Park et al., 2025). The second is *image quality degradation*, a form of reward hacking where the model over-optimizes for the reward signal (e.g., an aesthetic score) at the expense of general fidelity, producing artifacts and unrealistic outputs (Wang & Yu, 2025). In this work, we use "model collapse" as the umbrella term to refer to this overall process where both sample diversity and fidelity are compromised.

## 2 PCPO: PROPORTIONATE CREDIT ASSIGNMENT POLICY OPTIMIZATION

### 2.1 PRELIMINARIES

**Diffusion and Flow Matching.** Conditional diffusion probabilistic models (Ho et al., 2020) learn to create data by reversing a Markovian forward process that gradually adds Gaussian noise to a clean sample $\mathbf{x}_0$. The model, $\boldsymbol{\varepsilon}_\theta(\mathbf{x}_t, t, c)$, is trained to predict the noise $\boldsymbol{\epsilon}$ added at an intermediate state $\mathbf{x}_t$, typically by minimizing a weighted mean-squared error objective. Flow matching models (Lipman et al., 2023) simplify this process by learning the velocity $\mathbf{u}_\theta(\mathbf{x}_t, t, c)$, which is typically applied to follow the straight-line path between Gaussian noise and the data sample (Liu et al., 2023). This allows for efficient generation via solving a deterministic ordinary differential equation (ODE).

**Policy Gradient Alignment.** We frame the image generation process as a Markov Decision Process (MDP) (Bellman, 1957) following the formulation of Black et al. (2024). The $T$-step reverse process is defined by states $s_t = (\mathbf{x}_t, t, c)$ and actions $a_t = \mathbf{x}_{t-1}$, conditioned on a prompt $c$. A terminal reward $r(\mathbf{x}_0, c)$ is assigned at the final step. To improve sample efficiency, PPO performs multiple optimization steps on trajectories from an older policy $\pi_{\theta_{\text{old}}}$, clipping the importance sampling ratio $\rho_t(\theta) := p_\theta^{(t)}/p_{\theta_{\text{old}}}^{(t)}$ to stabilize updates (Fan et al., 2023; Schulman et al., 2017):

$$\mathcal{L}_{\text{PPO}}(\theta) := \mathbb{E}_{\tau \sim p_{\theta_{\text{old}}}} \left[ \sum_{t=1}^T \max\left( -\rho_t A, -\text{clip}_\xi(\rho_t) A \right) \right], \tag{1}$$

where $A$ is the normalized terminal reward and $\text{clip}_\xi(\rho_t) := \text{clip}(\rho_t, 1 + \xi, 1 - \xi)$ for clipping threshold $\xi$. The state-of-the-art GRPO (Shao et al., 2024) employs an analogous objective, enhancing stability by performing group-relative reward normalization to calculate the advantage, i.e. $\hat{A}^i = (r^i - \mu_G)/\sigma_G$, from a group of $G$ samples. Since our contribution, PCPO, focuses on modifying the policy ratio $\rho_t$ and its underlying credit assignment—mechanisms common to both frameworks—we proceed using the simpler PPO notation for our derivation. We omit the KL penalty term for simplicity, following prior work (Black et al., 2024; Xue et al., 2025).

### 2.2 PCPO DERIVATION

**PCPO for Diffusion Models.** Following Huang et al. (2024), we note that the gradient of Eq. (1) is equivalent to that of a hinge loss,

$$\mathcal{L}_{\text{hinge}} := \mathbb{E}[\sum_t \max\{0, \xi|A| - A(\rho_t - 1)\}]. \tag{2}$$

We stabilize this objective by replacing the numerically unstable term $\rho_t - 1$ with the more robust $\log \rho_t$. This choice is justified in two ways. First, under the hinge loss interpretation (Huang et al., 2024), this term acts as a swappable "classifier," allowing us to substitute different functions while maintaining the core mechanism. Second, it is a sound Taylor approximation for small policy updates

($\log \rho_t \approx \rho_t - 1$), a condition enforced by the small clipping range in our experiments; we empirically confirmed that this approximation error never exceeded 1.2% during training. More justification is provided in Appendix D. This leads to our stable log-hinge objective:

$$\mathcal{L}_{\text{PCPO-base}}(\theta) := \mathbb{E}\left[\sum_{t=1}^{T} \max\{0,\ \xi|A| - A\log\rho_t\}\right].\tag{3}$$

The core issue of disproportionate credit, however, lies within the $\log\rho_t$ term itself. We decompose this term in the following proposition, with the full derivation provided in Appendix A.1.

**Proposition 1.** *For a DDIM sampling schedule, the log policy ratio $\log\rho_t$ is given by:*

$$\log\rho_t = -\left[w(t)(\hat{\varepsilon}_\theta^{(t)} - \hat{\varepsilon}_{old}^{(t)})\cdot\boldsymbol{\epsilon}_{old}^{(t)} + \tfrac{1}{2}\big\|w(t)(\hat{\varepsilon}_\theta^{(t)} - \hat{\varepsilon}_{old}^{(t)})\big\|^2\right],\qquad w(t) = \frac{C(t)}{\sigma_t},\tag{4}$$

*where*

$$C(t) = \frac{\sqrt{1-\bar{\alpha}_t}}{\sqrt{\alpha_t}} - \sqrt{1 - \bar{\alpha}_{t-1} - \sigma_t^2}\ >\ 0.$$

Here, $\hat{\varepsilon}$ denotes the denoiser's noise prediction[1], whereas $\boldsymbol{\epsilon}_{\text{old}}$ is the Gaussian noise sampled during reverse sampling under the old policy. Substituting Eq. (4) into Eq. (3) transforms the PPO objective into an equivalent $\varepsilon$-matching loss

$$\mathcal{L}_{\varepsilon\text{-matching}}(\theta) = \mathbb{E}\left[\sum_{t=1}^{T} \max\left\{0,\ \xi|A| + A\mathcal{D}(w(t),\hat{\varepsilon}_\theta^{(t)},\hat{\varepsilon}_{\text{old}}^{(t)},\boldsymbol{\epsilon}_{\text{old}}^{(t)})\right\}\right]\tag{5}$$

where

$$\mathcal{D}(w(t),\hat{\varepsilon}_\theta^{(t)},\hat{\varepsilon}_{\text{old}}^{(t)},\boldsymbol{\epsilon}_{\text{old}}^{(t)}) := w(t)(\hat{\varepsilon}_\theta^{(t)} - \hat{\varepsilon}_{\text{old}}^{(t)})\cdot\boldsymbol{\epsilon}_{\text{old}}^{(t)} + \tfrac{1}{2}\big\|w(t)(\hat{\varepsilon}_\theta^{(t)} - \hat{\varepsilon}_{\text{old}}^{(t)})\big\|^2.\tag{6}$$

This decomposition reveals that the gradient contribution of each timestep is scaled by a native weight $w(t)$ that is highly non-uniform, spanning orders of magnitude (Figure 2a). This variance is a primary source of training instability, as it both causes gradients from different timesteps to be scaled inconsistently and leads to the most amplified gradients being clipped disproportionately often.

We argue that for proper credit assignment, these weights should be uniform. This principle is justified by a direct analogy to the foundational REINFORCE policy gradient algorithm (Sutton & Barto, 2018; Williams, 1992). In that framework, parameter updates are proportional to the *eligibility vector* (the policy gradient term), which is scaled by each action's contribution (often assumed to be uniform). Our analysis (see Appendix E) shows that the diffusion sampler's gradient formulation is analogous, but with a critical distinction: it scales this "eligibility vector" by a non-uniform, arbitrary weight $w(t)$. This $w(t)$ is an artifact of the sampler's mathematics, not a deliberate credit assignment strategy. This native scaling introduces high variance by weighting the credit $A$ based on the noise schedule rather than the step's actual importance.

PCPO restores credit assignment proportional to the integration interval by re-engineering the DDIM variance schedule, $\tilde{\sigma}_t$, to produce a constant weight, $w(t) = w^\star$, for all timesteps. To do this, we use the definition of the weight from Proposition 1, $w(t) = C(t)/\sigma_t$. For each timestep $t$, we set the weight on the left-hand side to our target constant $w^\star$. With the standard DDIM schedule ($\alpha_t$) fixed on the right-hand side, the only free parameter remaining in the equation is the variance $\sigma_t$. We can therefore solve for the precise value of $\sigma_t$ at each step that yields our desired constant weight. To ensure a fair comparison and isolate the effect of this uniform weighting, we rescale $w^\star$ to match the mean of the original, non-uniform weights (see Figure 2(a, b)).

**PCPO for Flow Models.** Applying policy gradients to flow models requires introducing stochasticity via a reverse SDE with the same marginal probability densities as the original ODE (Liu et al., 2025; Xue et al., 2025):

$$d\mathbf{x}_t = \left(\mathbf{u}_t - \frac{1}{2}\sigma_t^2\nabla\log p_t(\mathbf{x}_t)\right)dt + \sigma_t d\mathbf{w}.\tag{7}$$

---

[1] Unless otherwise specified, $\hat{\varepsilon}$ is shorthand notation for noise prediction with classifier-free guidance (Ho & Salimans, 2021): $(1 - w_{\text{CFG}})\hat{\varepsilon}(\cdot,\emptyset) + w_{\text{CFG}}\hat{\varepsilon}(\cdot,c)$.

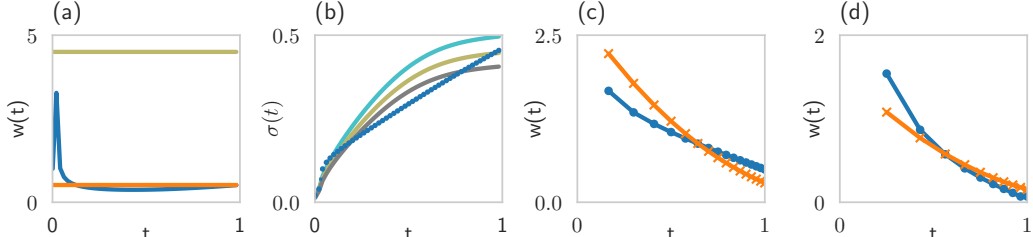

Figure 2: **Weight rescaling by PCPO. DDIM Sampler:** (a) Volatile native weights $w(t)$ (blue) are replaced with uniform, rescaled weight (orange). (b) This is achieved by computing a new variance signal $\tilde{\sigma}_t$ that remains close to the original (corresponding to $w^\star = 4.5$ (olive)), then rescaling. Light blue corresponds to $w^\star = 4.0$, gray to $w^\star = 5.0$. **SDE Sampler:** Native (blue) and rescaled (orange) weights for (c) DanceGRPO SDE, (d) Flow-GRPO SDE.

Integrating this SDE using first-order Euler-Maruyama discretization allows for trajectory sampling, where the log policy ratio for a single step can be simplified to a form analogous to Eq. (4):

$$\log \rho_{t_i} = -\Big[w(t_i)(\mathbf{u}_\theta - \mathbf{u}_{\text{old}}) \cdot \boldsymbol{\epsilon}_{\text{old}}^{(t_i)} + \frac{1}{2}\|w(t_i)(\mathbf{u}_\theta - \mathbf{u}_{\text{old}})\|^2\Big], \quad w(t_i) = \frac{\sqrt{\Delta t_i}}{\sigma_{t_i}}\Big(1 + \frac{(1-t_i)\sigma_{t_i}^2}{2t_i}\Big). \tag{8}$$

The challenge of disproportionate credit is particularly complex in modern flow-matching models. We illustrate this issue using the DanceGRPO SDE, where $\sigma_{t_i}$ is a constant, $\eta$ (Xue et al., 2025). For high-resolution synthesis, the timestep shifting technique (Esser et al., 2024) creates non-uniform integration intervals $\Delta t_i$, making native weights highly non-proportional to the interval length ($w(t_i) \not\propto \sqrt{\Delta t_i}$). While we restored proportionality in diffusion models with a minor, non-degrading adjustment to its variance schedule, an analogous modification for flow models requires drastic changes to the variance schedule or the timestep shifting strategy. Both options are problematic, as it significantly deviates from the original, well-optimized sampling procedure (Esser et al., 2024; Xue et al., 2025). Therefore, PCPO takes a different approach for flow models: it enforces proportionality by directly reweighting the training objective, as defined in the following proposition.

**Proposition 2.** *For a flow matching SDE in the form of Eq. (7), the weight schedule $w(t_i)$ that ensures credit is proportional to the integration interval $\Delta t_i$ is given by:*

$$w(t_i) = \zeta \Delta t_i, \qquad \zeta = \sum_{i=1}^{N} \frac{\sqrt{\Delta t_i}}{\sigma_{t_i}}\Big(1 + \frac{(1-t_i)\sigma_{t_i}^2}{2t_i}\Big). \tag{9}$$

The proof for Proposition 2 is in Appendix A.3.

Figure 2(c, d) compares the default (vanilla) and our improved (proportional) timestep weights for both the DanceGRPO and Flow-GRPO SDEs. The plots are generated using hyperparameters from the original works: $N = 16, \eta = 0.3$ for DanceGRPO (Xue et al., 2025), and $N = 10, \eta = 0.7$ for Flow-GRPO (Liu et al., 2025). Corresponding values of $\zeta$ are 13.343 and 4.315, respectively. For the Flow-GRPO visualization, we follow its official implementation and approximate the final weight $w_N$ (at $t = 1$) to avoid a divide-by-zero error in the variance schedule, $\sigma_t = \eta\sqrt{t/(1-t)}$.

## 3 EXPERIMENTS

### 3.1 METHODS

Our main analysis focuses on applying PCPO to two policy gradient frameworks: DDPO (Black et al., 2024) on Stable Diffusion 1.5 (SD1.5; Rombach et al. (2022)), and the state-of-the-art Dance-GRPO (Xue et al., 2025) on both Stable Diffusion 1.4 (SD1.4) and the FLUX.1-dev (FLUX) flow-matching model (Labs, 2024). We train DDPO on two reward models: Aesthetics (Schuhmann, 2022) and BERTScore (Zhang* et al., 2020), and DanceGRPO on HPSv2.1 (Wu et al., 2023). Our evaluation is twofold: we first analyze *training dynamics* by tracking reward acceleration and clipping fractions throughout the learning trajectory. We then assess *sample quality* at matched reward levels using

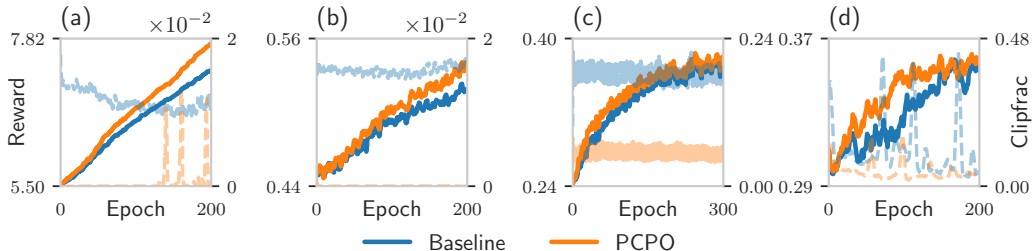

Figure 3: Reward and clipping fraction traces for PCPO (orange) vs. baselines (blue): (a) DDPO, Aesthetics, (b) DDPO, BERTScore, (c) DanceGRPO (SD1.4), HPSv2.1, (d) DanceGRPO (FLUX), HPSv2.1.

Table 1: Training Efficiency of Baseline vs. PCPO. Speedup in epochs translate directly to wall-clock time savings (see Figure 9).

| Baseline | Reward | Target Level | Epochs$_{Baseline}$ | Epochs$_{PCPO}$ | Speedup |
|---|---|---|---|---|---|
| DDPO | Aesthetics | 6.90 | 147 | **118** | **24.6%** |
| DDPO | BERTScore | 0.52 | 191 | **146** | **30.8%** |
| DanceGRPO (SD1.4) | HPS | 0.370 | 236 | **188** | **25.5%** |
| DanceGRPO (FLUX) | HPS | 0.360 | 209 | **148** | **41.2%** |

Table 2: Raw scores for (a) DDPO (Aesthetics) and (b) DanceGRPO (HPS) experiments. PCPO was put at a disadvantage for the DanceGRPO evaluation. Statistically significant differences in **bold**.

| | (a) | | | | | | (b) | | | |
|---|---|---|---|---|---|---|---|---|---|---|
| **Batch** | **Method** | **FID** | **FD$_{DINO}$** | **IS*** | **LPIPS** | **Model** | **Method** | **FID** | **FD$_{DINO}$** | **IS*** | **LPIPS** |
| 256 | Baseline | 31.72 | 473.17 | 26.35 | 0.6208 | SD1.4 | Baseline | 90.34 | 1078.42 | 7.61 | 0.4948 |
| | PCPO | **27.86** | 461.69 | **24.12** | 0.6262 | | PCPO | **84.74** | 1035.45 | 7.50 | 0.4894 |
| 512 | Baseline | 24.09 | 451.19 | 25.67 | 0.6321 | FLUX | Baseline | 46.23 | 539.83 | 12.66 | 0.5736 |
| | PCPO | **22.06** | 391.30 | **25.65** | **0.6525** | | PCPO | **40.38** | **438.88** | **11.90** | 0.5708 |

Fréchet Inception Distance (FID) (Heusel et al., 2018), Fréchet Distance DINOv2 (FD$_{DINO}$) (Stein et al., 2023), Inception Score (IS) (Salimans et al., 2016), and LPIPS Diversity (Zhang et al., 2018). To account for per-prompt variance in these quality metrics, we validate our findings using a Linear Mixed Model (LMM). All main results are from experiments using main configurations from Table 7 in Appendix B, with the exception of the half-sized batch configurations used for LMM analysis. All qualitative results are also generated from experiments using main configurations, with the exception of Figure 7(b).

To further validate PCPO's robustness, we benchmark performance on unseen prompts from the MSCOCO-2017 validation (5K) (Lin et al., 2014) and MJHQ-30K (Li et al., 2024) datasets, evaluating on a diverse suite of alignment metrics: HPSv2.1, Aesthetics, CLIPScore (Hessel et al., 2021), PickScore (Kirstain et al., 2023), and ImageReward (Xu et al., 2023). Next, we test PCPO's generalizability by applying it to the Flow-GRPO SDE framework (Liu et al., 2025) on the SD3.5-M model (Esser et al., 2024)—a substantially different architecture and training setup that uses different rewards (OCR (Chen et al., 2023), PickScore) and an auxiliary KL divergence penalty. Full experimental details are provided in Appendix B.

## 3.2 RESULTS

**PCPO Improves Training Efficiency and Stability.** PCPO's core principle of proportionate credit assignment translates directly to enhanced training stability. As shown in Figure 3, PCPO consistently maintains a lower and more stable clipping fraction than the baselines. This stability is the key to its faster convergence, leading to substantial training acceleration across all experimental settings (Table 1). A detailed breakdown of speedups is available in Table 10 in Appendix D.

**PCPO Mitigates Model Collapse.** The stability from PCPO translates to significant improvements in fidelity and diversity. Specifically, for the DDPO experiments, PCPO achieves a statistically

significant improvement in sample fidelity (FID). While $FD_{DINO}$ showed similar trends, the effect was not statistically significant ($p = 0.247$). We attribute this lack of significance to limitations of the base model or task setup, especially since the $FD_{DINO}$ metric also failed to register a significant effect for batch size ($p = 0.468$). This suggests the metric may be insensitive to improvements within this specific experimental context, possibly due to simplicity of prompts or limitations of the base model's capacity.

The effect on LPIPS diversity is more nuanced: while the main effect of PCPO alone was not statistically significant, the LMM analysis reveals a strong, positive interaction between the algorithm and batch size ($\beta_{int} = 0.016, p = 0.008$). This indicates PCPO's benefit to diversity becomes prominent when synergizing with larger batch sizes, which on their own were found to significantly increase diversity ($\beta_{batch} = 0.010, p = 0.016$).

The most compelling evidence for PCPO's role in mitigating model collapse, however, comes from the Inception Score (IS) analysis. We acknowledge the common interpretation that, given similar FID, a higher IS can indicate better quality. However, this metric's behavior can be task-dependent, and it is known that non-diverse models can "artificially achieve high IS" (Sadat et al., 2024). To determine the correct interpretation for our specific task, we first established an empirical ground truth. We found that increasing the batch size—a known technique to reduce model collapse by preserving data diversity (Shumailov et al., 2024)—causes a statistically significant *decrease* in IS ($\beta_{batch} = -0.266, p = 0.011$).

Table 3: LMM analysis for DDPO (Aesthetics).

| Effect | Metric | Coeff. ($\beta$) | p-value |
|---|---|---|---|
| Algorithm | FID | **-7.750** | **0.047** |
| | $FD_{DINO}$ | -52.831 | 0.247 |
| | IS* | **-0.241** | **0.021** |
| | LPIPS | 0.004 | 0.401 |
| Batch Size | FID | **-13.509** | **0.001** |
| | $FD_{DINO}$ | -46.811 | 0.468 |
| | IS* | **-0.266** | **0.011** |
| | LPIPS | **0.010** | **0.016** |
| Interaction | FID | 4.053 | 0.463 |
| | $FD_{DINO}$ | -55.720 | 0.222 |
| | IS* | 0.181 | 0.221 |
| | LPIPS | **0.016** | **0.008** |

This finding strongly suggests that, in this context, a high IS is not an indicator of quality but rather a pathological artifact of mode collapse, rewarding low-diversity, high-confidence outputs. We therefore treat IS for our task as a metric to be minimized, given lower or comparable FID. With this understanding, PCPO's statistically significant reduction in IS ($\beta_{alg} = -0.241, p = 0.021$) is not a sign of degradation, but strong evidence that it achieves a desirable stabilizing effect, similar to using a larger batch.

Table 4: LMM analysis for DanceGRPO (HPS).

| Model | Metric | Coeff. ($\beta$) | p-value |
|---|---|---|---|
| SD1.4 | FID ($\downarrow$) | **-6.730** | **0.026** |
| | $FD_{DINO}$ ($\downarrow$) | -64.482 | 0.128 |
| | IS* ($\downarrow$) | -0.029 | 0.499 |
| | LPIPS ($\uparrow$) | -0.005 | 0.102 |
| FLUX | FID ($\downarrow$) | **-13.884** | **0.001** |
| | $FD_{DINO}$ ($\downarrow$) | **-175.592** | **0.001** |
| | IS* ($\downarrow$) | **-0.184** | **0.014** |
| | LPIPS ($\uparrow$) | -0.002 | 0.629 |

PCPO's fidelity improvements (FID, $FD_{DINO}$) also hold in the DanceGRPO experiments. This is particularly notable because the comparison was handicapped; due to noisy reward trajectories, we evaluated PCPO at a higher-reward checkpoint than the baseline, a state that typically harms FID. Nonetheless, PCPO still delivered significantly better FID for both models. This result, along with a lower IS for the FLUX model, reinforces that PCPO effectively mitigates model collapse. Conversely, no significant effect on LPIPS was observed for the DanceGRPO experiments. We hypothesize this is due to two confounding factors: the batch sizes may have been too small to activate the synergistic effect, and the comparison was made at a checkpoint where PCPO had already achieved a higher reward, potentially masking underlying diversity improvements.

Qualitatively, PCPO avoids the severe model collapse that affects the baselines, producing clear and diverse images instead of the blurry, repetitive outputs characteristic of unstable training (Figure 1). Further comparisons showing PCPO's superior visual fidelity throughout training are available in Appendix H. To validate these gains with human perception, we conducted a formal preference study. To ensure a fair comparison, we evaluated PCPO (epoch 120) against the baseline at two reward-bracketing checkpoints (epochs 180 and 240). The results were decisive: human evaluators robustly preferred PCPO's outputs over the DanceGRPO baseline in all categories (Figure 4). This

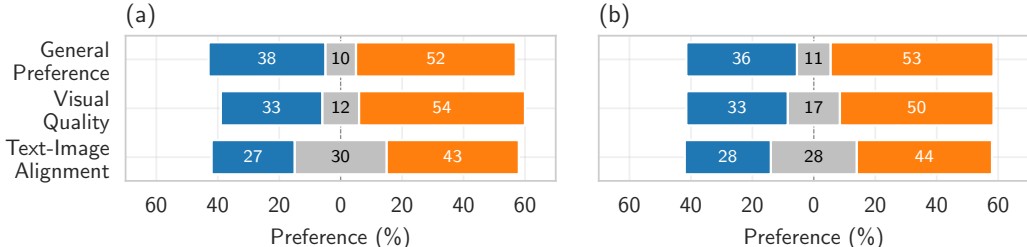

Figure 4: Human evaluation results for DanceGRPO (blue) vs. PCPO (orange) on FLUX. To ensure a fair comparison that accounts for faster convergence, our model (120 epochs) was evaluated against the baseline at two reward-bracketing checkpoints: (a) 180 epochs and (b) 240 epochs.

Table 5: Human preference alignment metrics on 5K unseen prompts from (a) MSCOCO-2017 Validation set, (b) MJHQ-30K. PCPO outperforms DanceGRPO on the trained reward (HPSv2.1) and unseen metrics, even at an earlier epoch (FLUX), confirming less reward hacking.

| Model | Method (Epoch) | HPSv2.1 | Aesthetic | CLIPScore | PickScore | ImgRwd |
|-------|----------------|---------|-----------|-----------|-----------|--------|
| | **(a) MSCOCO-2017 Val** | | | | | |
| SD1.4 | Base Model | 0.252 | 5.18 | **0.365** | 21.53 | 0.17 |
| | DanceGRPO (E200) | 0.317 | **5.69** | 0.357 | 22.13 | 0.87 |
| | PCPO (E200) | **0.326** | **5.69** | **0.365** | **22.26** | **0.90** |
| FLUX | Base Model | 0.293 | 5.70 | **0.382** | 22.94 | 0.95 |
| | DanceGRPO (E180) | 0.330 | 6.15 | 0.373 | 23.08 | 1.07 |
| | PCPO (E120) | **0.337** | **6.21** | 0.369 | **23.12** | **1.14** |
| | **(b) MJHQ** | | | | | |
| SD1.4 | Base Model | 0.247 | 5.56 | 0.370 | 19.84 | 0.17 |
| | DanceGRPO (E200) | 0.347 | 6.14 | 0.374 | 20.95 | 1.12 |
| | PCPO (E200) | **0.353** | **6.18** | **0.379** | **21.14** | **1.16** |
| FLUX | Base Model | 0.306 | 6.37 | **0.398** | 21.96 | 1.14 |
| | DanceGRPO (E180) | 0.345 | **6.56** | 0.387 | 22.14 | 1.27 |
| | PCPO (E120) | **0.350** | **6.56** | **0.398** | **22.28** | **1.32** |

preference was strong even on mobile devices where the baseline's subtle artifacts (Figure 20) were less visible, suggesting the true performance gap may be even wider.

**Evaluation with Unseen Prompts on Diverse Reward Metrics.** We evaluated checkpoints at matched reward levels from our HPSv2.1-trained DanceGRPO experiments. We test generalization on simpler prompts from the MSCOCO-2017 validation set and more complex prompts from the MJHQ-30K dataset. The results, presented in Table 5(a, b), demonstrate that PCPO not only excels on the metric it was trained on (HPSv2.1) but also outperforms the baseline across a wide suite of metrics. For SD1.4, PCPO outperforms DanceGRPO across all metrics at an equivalent training point (200 epochs). For the FLUX model, PCPO at 120 epochs outperforms the DanceGRPO baseline at 180 epochs. This confirms that PCPO converges substantially faster to a more generalizable policy. Notably, PCPO maintains text-image alignment (CLIPScore) better than the baseline, suggesting it is less prone to reward hacking.

**Generalization.** To test generalizability, we applied PCPO in a significantly different training regime using the SD3.5-M model with the Flow-GRPO framework. This setup involved a distinct noise schedule, different rewards (OCR, PickScore), an auxiliary KL penalty, and a separate prompt dataset (see Appendix B). Despite these substantial changes, PCPO's benefits remained clear. As shown in Figure 5, PCPO consistently outperformed the baseline, achieving a higher reward and lower (better) KL divergence, while maintaining significantly less clipping. This confirms PCPO's stabilizing properties are robust across different models, noise schedules, and training configurations.

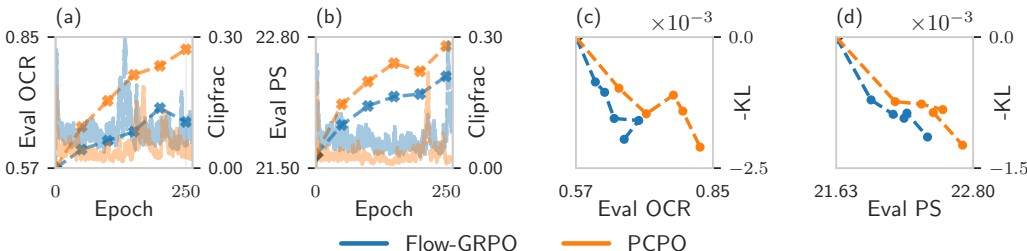

Figure 5: PCPO demonstrates robust generalizability on SD3.5-M using the Flow-GRPO framework. PCPO (orange) consistently outperforms the baseline (blue), achieving a higher reward, lower KL divergence, and a lower clipping fraction. Plots explained in depth in Appendix B.3.4.

## 3.3 ABLATION STUDY

**PCPO Component Analysis.** Our ablation study dissects the contribution of each PCPO component by sequentially adding them to the DDPO (Aesthetics) baseline. Figure 6 shows this progression—from DDPO (blue), through the $\log \rho_t$ objective (green) and $\varepsilon$-matching (red), to our final proportional weighting (orange)—yields a steady reduction in clipping. While each component contributes to stability, only the full PCPO framework achieves a zero on-policy clipping ratio, a theoretical result confirming its superior design. This final step of adding proportional weighting was also the most critical factor for accelerating training.

**PCPO vs. Heuristic Acceleration Methods.** To validate our principled approach, we contrast PCPO against two heuristic acceleration strategies. First, we compare against *timestep subsampling*, a heuristic where policy updates use only 50% of timesteps. This reduces wall-clock time per epoch to just 58% of the standard training, but as Figure 7(b) shows, the speedup comes at a significant cost to final image quality.

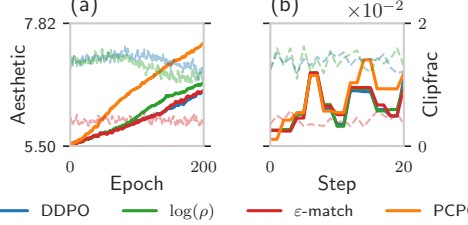

Figure 6: Ablation study of sequentially adding PCPO's components. Results show rewards and clipping fractions over (a) 200 epochs, (b) a close-up of the initial 20 steps (unsmoothed).

Second, we contrast our proportionality principle with a heuristic inspired by the empirical observations of concurrent work (He et al., 2025): prioritizing high-noise timesteps due to their higher reward variance. The DanceGRPO SDE provides a decisive test case, as its native weights ($\propto \sqrt{\Delta t_i}$) already disproportionately favor the short, high-noise integration steps, unlike the Flow-GRPO SDE used by TempFlow-GRPO (Figure 2(c, d)). The empirical heuristic would suggest amplifying these weights even further. In direct opposition, our proportionality principle requires reweighting in the opposite direction. Our main experiments confirm that applying our principle remarkably accelerates training. Conversely, further emphasizing high-noise steps with a uniform weighting schedule ultimately harms performance compared to the baseline (Figure 7(d)). Together, these results highlight the advantage of our principled approach: PCPO consistently achieves significant training acceleration *without* the degradation in sample quality that often accompanies simple heuristic methods.

## 4 DISCUSSION

Our results strongly suggest that PCPO effectively mitigates model collapse. This phenomenon, where models degrade when trained on recursively generated data, arises from several error sources accumulating over time. The primary driver is the *statistical approximation error* from using finite training samples, which causes a progressive loss of the data distribution's tails (Shumailov et al., 2024). Standard policy gradient methods aggravate this issue by aggressively clipping and discarding these crucial tail-end samples. While the most direct method to reduce this specific error is to increase the batch size, this approach incurs significant computational overhead and does not address other contributing factors, such as numerical precision errors, which are also implicated in model collapse.

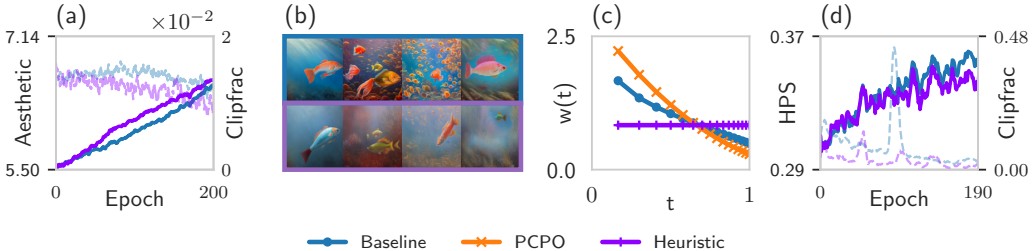

Figure 7: (a) The timestep subsampling method (purple) yields similar reward gain per epoch, which translates to a significant reduction in wall-clock time per reward gain (58% of the DDPO baseline, blue). (b) But this speedup degrades sample quality and diversity, shown here for the prompt "fish" using 4 consecutive seeds at a matched reward level (Aesthetics = 6.90). Top row: DDPO, bottom row: DDPO + timestep subsampling. (c) Purple: A uniform reweighting scheme that emphasizes high-noise ($t \approx 1$) timesteps. (d) The uniform scheme shows a marginal initial speedup but is ultimately outperformed by the vanilla $\propto \sqrt{\Delta t_i}$ schedule of DanceGRPO (blue).

PCPO offers a more comprehensive and efficient solution by targeting multiple error sources. It directly counters the statistical error by preserving more tail-end data through drastically reduced clipping. Simultaneously, its $\log \rho$ formulation addresses the secondary issue of numerical precision errors. Our LMM analysis confirms that the combined result of these targeted corrections is a stabilizing effect paralleling that of doubling the batch size (Table 3). PCPO therefore provides the benefits of larger batch training without the associated computational overhead, offering an efficient, multi-faceted defense against model collapse.

# 5 CONCLUSION AND FUTURE WORK

In this work, we addressed the critical instability in policy-gradient based alignment for T2I models, identifying the root cause as *disproportionate credit assignment*. Our proposed framework, *Proportionate Credit Policy Optimization (PCPO)*, corrects this fundamental flaw by ensuring the feedback signal from each timestep is proportional to its contribution. This principled correction was shown to accelerate convergence and improve sample quality, achieving state-of-the-art performance that surpasses strong baselines, including DanceGRPO.

This work's focus on providing a stable, proportional feedback signal for alignment opens several exciting avenues for future research. A promising direction is to explore the synergy between PCPO's credit proportionality and other stabilization techniques, such as dynamic clipping, temporal localization, and KL regularization. A particularly interesting avenue relates to *gradient clipping*. While PCPO reduces the need for PPO and GRPO's internal *timestep clipping*, the training of flow models often still relies on downstream *gradient clipping* to prevent divergence. Interestingly, aggressive gradient clipping has been observed to destabilize training (Xue & xingzhejun, 2025). Therefore, a deeper investigation into the underlying mechanics that produce these large, unstable gradients presents a fruitful research direction, potentially leading to alignment methods that are stable by design.

**Reproducibility Statement.** For every proposition and mathematical derivation, we provide proofs in Appendix A. Hyperparameters and resources required to reproduce experiments are stated in Appendix B. Code is available at `https://github.com/jaylee2000/pcpo/`.

**Acknowledgement.** This research was supported by the AI Computing Infrastructure Enhancement (GPU Rental Support) User Support Program funded by the Ministry of Science and ICT (MSIT), Republic of Korea (RQT-25-120217). This work was supported by the National Research Foundation of Korea under Grant RS-2024-00336454. This work was supported by the Institute of Information & Communications Technology Planning & Evaluation(IITP) grant funded by the Korea government(MSIT) (RS-2025-02304967, AI Star Fellowship(KAIST)). This work was supported by Institute for Information & communications Technology Planning & Evaluation(IITP) grant funded by the Korea government(MSIT) (RS-2019-II190075, Artificial Intelligence Graduate School Program(KAIST)).

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

## A  DERIVATIONS FOR PCPO OBJECTIVE

### A.1  PROOF OF PROPOSITION 1

**Proposition 1.** *For a DDIM sampling schedule, the log policy ratio* $\log \rho_t$ *is given by:*

$$\log \rho_t = -\Big[w(t)(\hat{\varepsilon}_\theta^{(t)} - \hat{\varepsilon}_{old}^{(t)}) \cdot \epsilon_{old}^{(t)} + \tfrac{1}{2}\big\|w(t)(\hat{\varepsilon}_\theta^{(t)} - \hat{\varepsilon}_{old}^{(t)})\big\|^2\Big], \qquad w(t) = \frac{C(t)}{\sigma_t}, \qquad (4)$$

*where*

$$C(t) = \frac{\sqrt{1 - \bar{\alpha}_t}}{\sqrt{\alpha_t}} - \sqrt{1 - \bar{\alpha}_{t-1} - \sigma_t^2} \; > \; 0.$$

*Proof.*

$$\log \rho_t = \log \frac{\mathcal{N}\Big(\mathbf{x}_{t-1}; \hat{\boldsymbol{\mu}}_\theta^{(t-1)}(x_t), \sigma_t^2 \mathbf{I}\Big)}{\mathcal{N}\Big(\mathbf{x}_{t-1}; \hat{\boldsymbol{\mu}}_{old}^{(t-1)}(x_t), \sigma_t^2 \mathbf{I}\Big)}$$

$$= -\frac{1}{2\sigma_t^2}\Big(\big\|\mathbf{x}_{t-1} - \hat{\boldsymbol{\mu}}_\theta^{(t-1)}(x_t)\big\|^2 - \big\|\mathbf{x}_{t-1} - \hat{\boldsymbol{\mu}}_{old}^{(t-1)}(x_t)\big\|^2\Big)$$

*where*

$$\hat{\boldsymbol{\mu}}_\theta^{(t-1)}(\mathbf{x}_t) := \frac{\mathbf{x}_t - \sqrt{1 - \bar{\alpha}_t}\,\hat{\varepsilon}_\theta^{(t)}(\mathbf{x}_t)}{\sqrt{\alpha_t}} + \sqrt{1 - \bar{\alpha}_{t-1} - \sigma_t^2}\,\hat{\varepsilon}_\theta^{(t)}(\mathbf{x}_t).$$

*and* $\mathbf{x}_{t-1} = \hat{\boldsymbol{\mu}}_{old}^{(t-1)}(\mathbf{x}_t) + \sigma_t \epsilon_{old}^{(t)} \sim \pi_{old}$. *Substituting this into the equation above yields*

$$-\frac{1}{2\sigma_t^2}\Big(\big\|\mathbf{x}_{t-1} - \hat{\boldsymbol{\mu}}_\theta^{(t-1)}(\mathbf{x}_t)\big\|^2 - \big\|\mathbf{x}_{t-1} - \hat{\boldsymbol{\mu}}_{old}^{(t-1)}(\mathbf{x}_t)\big\|^2\Big)$$

$$= -\frac{1}{2\sigma_t^2}\Big[\|\hat{\boldsymbol{\mu}}_{old}^{(t-1)}(\mathbf{x}_t) - \hat{\boldsymbol{\mu}}_\theta^{(t-1)}(\mathbf{x}_t) + \sigma_t \epsilon_{old}^{(t)}\|^2 - \|\sigma_t \epsilon_{old}^{(t)}\|^2\Big]$$

$$= -\frac{1}{2\sigma_t^2}\Big[\|C(t)(\hat{\varepsilon}_\theta^{(t)}(\mathbf{x}_t) - \hat{\varepsilon}_{old}^{(t)}(\mathbf{x}_t)) + \sigma_t \epsilon_{old}^{(t)}\|^2 - \|\sigma_t \epsilon_{old}^{(t)}\|^2\Big]$$

$$= -\frac{1}{2}\Big[\|w(t)(\hat{\varepsilon}_\theta^{(t)}(\mathbf{x}_t) - \hat{\varepsilon}_{old}^{(t)}(\mathbf{x}_t)) + \epsilon_{old}^{(t)}\|^2 - \|\epsilon_{old}^{(t)}\|^2\Big]$$

$$= -\Big[w(t)(\hat{\varepsilon}_\theta^{(t)}(\mathbf{x}_t) - \hat{\varepsilon}_{old}^{(t)}(\mathbf{x}_t)) \cdot \epsilon_{old}^{(t)} + \frac{1}{2}\|w(t)(\hat{\varepsilon}_\theta^{(t)}(\mathbf{x}_t) - \hat{\varepsilon}_{old}^{(t)}(\mathbf{x}_t))\|^2\Big].$$

$\square$

### A.2  PROOF OF EQ. (8)

Here, we provide a detailed derivation for the log policy ratio in the flow model setting, analogous to the proof for the diffusion model case.

We begin with the definition of the log policy ratio for a single timestep $t_i$, which is the log-likelihood ratio of the one-step transition probabilities. Both $p_\theta$ and $p_{old}$ define a Gaussian distribution for the next state $x_{t_i - \Delta t_i}$ given the current state $x_{t_i}$.

$$\log \rho_{t_i} = \log \frac{p_\theta(x_{t_i - \Delta t_i} \mid x_{t_i})}{p_{old}(x_{t_i - \Delta t_i} \mid x_{t_i})}$$

The SDE in Eq. (7) implies that the one-step transition is a sample from $\mathcal{N}(\hat{\boldsymbol{\mu}}_\theta^{(t_{i-1})}(\mathbf{x}_{t_i}), \boldsymbol{\Sigma}_{t_i})$, where the variance is $\boldsymbol{\Sigma}_{t_i} = \sigma_{t_i}^2(\Delta t_i)\mathbf{I}$ and the mean is $\hat{\boldsymbol{\mu}}_\theta^{(t_{i-1})}(\mathbf{x}_{t_i}) = \mathbf{x}_{t_i} - \mathbf{u}_\theta \Delta t_i + \frac{\sigma_{t_i}^2}{2}\mathbf{s}_\theta \Delta t_i$, where

$$\mathbf{s}_\theta = -\frac{\mathbf{x}_{t_i} - (1 - t_i)\hat{\mathbf{x}}_0}{t_i{}^2}, \qquad \hat{\mathbf{x}}_0 = \mathbf{x}_{t_i} - \mathbf{u}_\theta t_i. \qquad (10)$$

Using the formula for the log-ratio of two Gaussians with the same variance, we get:

$$\log \rho_{t_i} = -\frac{1}{2(\sigma_{t_i}\sqrt{\Delta t_i})^2} \left( \left\| \mathbf{x}_{t_i - \Delta t_i} - \hat{\boldsymbol{\mu}}_\theta^{(t_{i-1})}(\mathbf{x}_{t_i}) \right\|^2 - \left\| \mathbf{x}_{t_i - \Delta t_i} - \hat{\boldsymbol{\mu}}_{\text{old}}^{(t_{i-1})}(\mathbf{x}_{t_i}) \right\|^2 \right) \tag{11}$$

The next state $\mathbf{x}_{t_i - \Delta t_i}$ is sampled from the old policy, so $\mathbf{x}_{t_i - \Delta t_i} = \hat{\boldsymbol{\mu}}_{\text{old}}^{(t_{i-1})}(\mathbf{x}_{t_i}) + \sigma_{t_i}\sqrt{\Delta t_i}\,\boldsymbol{\epsilon}_{\text{old}}^{(t_i)}$. Substituting this into the equation yields:

$$\log \rho_{t_i} = -\frac{1}{2\sigma_{t_i}^2 \Delta t_i} \left( \left\| \hat{\boldsymbol{\mu}}_{\text{old}}^{(t_{i-1})}(\mathbf{x}_{t_i}) - \hat{\boldsymbol{\mu}}_\theta^{(t_{i-1})}(\mathbf{x}_{t_i}) + \sigma_{t_i}\sqrt{\Delta t_i}\,\boldsymbol{\epsilon}_{\text{old}}^{(t_i)} \right\|^2 - \left\| \sigma_{t_i}\sqrt{\Delta t_i}\,\boldsymbol{\epsilon}_{\text{old}}^{(t_i)} \right\|^2 \right) \tag{12}$$

Substituting Eq. (10) into the mean difference, we have:

$$\hat{\boldsymbol{\mu}}_{\text{old}}^{(t_{i-1})}(\mathbf{x}_{t_i}) - \hat{\boldsymbol{\mu}}_\theta^{(t_{i-1})}(\mathbf{x}_{t_i}) = \left( \mathbf{u}_\theta(x_{t_i}, t_i) - \mathbf{u}_{\text{old}}(x_{t_i}, t_i) \right) \Delta t_i \left( 1 + \frac{(1-t_i)\sigma_{t_i}^2}{2t_i} \right) \tag{13}$$

Substituting this back and expanding the squared norm, we get:

$$\log \rho_{t_i} = -\frac{1}{2\sigma_{t_i}^2 \Delta t_i} \left( \left\| (\mathbf{u}_\theta - \mathbf{u}_{\text{old}})\Delta t_i (1 + \frac{(1-t_i)\sigma_{t_i}^2}{2t_i}) + \sigma_{t_i}\sqrt{\Delta t_i}\,\boldsymbol{\epsilon}_{\text{old}}^{(t_i)} \right\|^2 - \left\| \sigma_{t_i}\sqrt{\Delta t_i}\boldsymbol{\epsilon}_{\text{old}}^{(t_i)} \right\|^2 \right)$$

$$= -\frac{1}{2\sigma_{t_i}^2 \Delta t_i} \left( \left\| (\mathbf{u}_\theta - \mathbf{u}_{\text{old}})\Delta t_i \left( 1 + \frac{(1-t_i)\sigma_{t_i}^2}{2t_i} \right) \right\|^2 \right.$$

$$\left. + 2(\mathbf{u}_\theta - \mathbf{u}_{\text{old}})\Delta t_i \left( 1 + \frac{(1-t_i)\sigma_{t_i}^2}{2t_i} \right) \cdot \left( \sigma_{t_i}\sqrt{\Delta t_i}\,\boldsymbol{\epsilon}_{\text{old}}^{(t_i)} \right) \right)$$

$$= -\frac{\Delta t_i}{2\sigma_{t_i}^2}(1 + \frac{(1-t_i)\sigma_{t_i}^2}{2t_i})^2 \|\mathbf{u}_\theta - \mathbf{u}_{\text{old}}\|^2 - \frac{\sqrt{\Delta t_i}}{\sigma_{t_i}}(1 + \frac{(1-t_i)\sigma_{t_i}^2}{2t_i})(\mathbf{u}_\theta - \mathbf{u}_{\text{old}}) \cdot \boldsymbol{\epsilon}_{\text{old}}^{(t_i)}$$

By defining the weight $w(t_i) = \frac{\sqrt{\Delta t_i}}{\sigma_{t_i}}(1 + \frac{(1-t_i)\sigma_{t_i}^2}{2t_i})$, we can rewrite the expression in a form analogous to the diffusion case:

$$\log \rho_{t_i} = -\left[ w(t_i)(\mathbf{u}_\theta - \mathbf{u}_{\text{old}}) \cdot \boldsymbol{\epsilon}_{\text{old}}^{(t_i)} + \frac{1}{2}\|w(t_i)(\mathbf{u}_\theta - \mathbf{u}_{\text{old}})\|^2 \right]. \tag{14}$$

This matches the structure of Eq. (8), confirming the derivation.

### A.3 PROOF OF PROPOSITION 2

**Proposition 2.** *For a flow matching SDE in the form of Eq. (7), the weight schedule $w(t_i)$ that ensures credit is proportional to the integration interval $\Delta t_i$ is given by:*

$$w(t_i) = \zeta \Delta t_i, \qquad \zeta = \sum_{i=1}^{N} \frac{\sqrt{\Delta t_i}}{\sigma_{t_i}}(1 + \frac{(1-t_i)\sigma_{t_i}^2}{2t_i}). \tag{9}$$

*Proof.* To derive $w(t_i) \propto \Delta t_i$ with the same mean weight as $w_{\text{orig}}(t_i)$, we define normalizing coefficient $\zeta$ such that:

$$\sum_{i=1}^{N} w_{\text{orig}}(t_i) = \sum_{i=1}^{N} w(t_i),$$

where

$$w_{\text{orig}}(t_i) = \frac{\sqrt{\Delta t_i}}{\sigma_{t_i}}(1 + \frac{(1-t_i)\sigma_{t_i}^2}{2t_i}), \quad w(t_i) = \zeta \Delta t_i.$$

Substituting this into the equation above yields $\zeta = \sum_{i=1}^{N} w_{\text{orig}}(t_i) = \sum_{i=1}^{N} \frac{\sqrt{\Delta t_i}}{\sigma_{t_i}}(1 + \frac{(1-t_i)\sigma_{t_i}^2}{2t_i})$.

$\square$

Table 6: Index of Training Configurations. This table links the configuration index (from Table 7) to its results in the paper. Main configurations in **bold**.

| Config | Brief Description | Figures | Tables |
|--------|------------------|---------|--------|
| 1 | DDPO Aes, Half Batch Size | 6, 7(a, b), 8(a) | 2(a), 3, 10 |
| 2 | **DDPO Aes** | 1(a), 3(a), 10(a), 11(a), 14, 15, 16, 17($\downarrow$) | 1, 2(a), 3, 10 |
| 3 | DDPO BERT, Half Batch Size | 8(b) | 2(a), 3, 10 |
| 4 | **DDPO BERT** | 3(b), 14, 17($\uparrow$) | 1, 2(a), 3, 10 |
| 5 | **DanceGRPO SD1.4** | 3(c), 13($\uparrow$), 18 | 1, 2(b), 4, 5, 10 |
| 6 | **DanceGRPO FLUX** | 1(b), 3(d), 4, 9(b), 10(b), 11(b), 13($\downarrow$), 19, 20 | 1, 2(b), 4, 5, 10 |
| 7 | DanceGRPO, DPO Comparison | | 9 |
| 8 | FlowGRPO SD3.5, OCR | 5(a, c) | |
| 9 | FlowGRPO SD3.5, PickScore | 5(b, d) | |
| 10 | DanceGRPO FLUX, Naive Acc. | 7(d) | |

## B  EXPERIMENT DETAILS

This section outlines the setup for all experiments, including training configurations, evaluation metrics, and computational resources.

### B.1  TRAINING SETUPS

We test PCPO across ten distinct training configurations, spanning three baseline algorithms, four base models, five reward functions, and five prompt datasets.

All experiments were conducted using fully online training, where each iteration generates fresh trajectories from the current policy. We specify important configurations and hyperparameters for each setup in Table 7. Apart from the learning rate, which was lowered from $3 \times 10^{-4}$ to $1 \times 10^{-4}$, and LoRA parameters for DDPO experiments, all hyperparameters follow that of the original codebases. For DDPO and Flow-GRPO, we used our memory-efficient reimplementations, following the logic of the official codebases.

#### B.1.1  TABLE 7 LEGEND

**Prompts.** "animal" refers to the `simple_animals` dataset from the DDPO codebase, consisting of 45 simple animal nouns (e.g., "cat", "dog", "fish"). "activity" refers to the `nouns_activities` dataset from the same codebase, consisting of 135 prompts combining 45 animal nouns with 3 activity verbs ("washing the dishes", "riding a bike", "playing chess"). "HPD" refers to the open-vocabulary prompts from the HPDv2 dataset used in Xue et al. (2025). "Pick" refers to the Pick-a-Pic dataset, and "Pick-SFW" refers to the safe-for-work (SFW) subset from the Flow-GRPO codebase. "OCR" refers to the OCR prompts from the OCR dataset.

**Reward.** "Aes", "BERT", "HPS", "CLIP", "Pick", and "OCR" are abbreviations for "Aesthetics", "BERTScore", "HPS-v2.1", "CLIP score", "PickScore", and "OCR" reward models, respectively.

**Timestep subsampling.** DanceGRPO(FLUX)'s default settings use the heuristic acceleration by subsampling 60% of timesteps for policy updates.

$\beta_{\mathbf{KL}}$. This is the coefficient for the KL penalty term for KL-regularized policy gradient algorithms, as in Equation (9) of Fan et al. (2023) or Equation (3) of Liu et al. (2025). An empty entry indicates no KL penalty used.

Table 7: All 10 Training Setups. Each column (1-10) indexes a unique setup. Configurations for main experiments are in **bold**.

| Parameter | 1 | 2 | 3 | 4 | 5 | 6 | 7 | 8 | 9 | 10 |
|---|---|---|---|---|---|---|---|---|---|---|
| *Framework & Model* | | | | | | | | | | |
| Baseline algorithm | DDPO | **DDPO** | DDPO | **DDPO** | **DanceGRPO** | **DanceGRPO** | DanceGRPO | Flow-GRPO | Flow-GRPO | DanceGRPO |
| Diffusion/Flow | D | **D** | D | **D** | **D** | **F** | D | F | F | F |
| Base model | SD1.5 | **SD1.5** | SD1.5 | **SD1.5** | **SD1.4** | **FLUX.1-dev** | SD1.5 | SD3.5-M | SD3.5-M | FLUX.1-dev |
| *Task* | | | | | | | | | | |
| Prompts | animal | **animal** | activity | **activity** | **HPD** | **HPD** | Pick | OCR | Pick-SFW | HPD |
| Reward | Aes | **Aes** | BERT | **BERT** | **HPS** | **HPS** | Pick | OCR | Pick | HPS |
| *Sampling Hyperparams* | | | | | | | | | | |
| Noise level (SDE) | | | | | | **$\eta = 0.3$** | | $\eta = 0.7$ | $\eta = 0.7$ | $\eta = 0.3$ |
| Timestep subsampling | | | | | | **0.6** | | | | 0.6 |
| Sampling steps | 50 | **50** | 50 | **50** | **50** | **12** | 50 | 10 | 10 | 16 |
| *Training Hyperparams* | | | | | | | | | | |
| $w_{\text{CFG}}$ | 5.0 | **5.0** | 5.0 | **5.0** | **5.0** | **1.0** | 5.0 | 4.5 | 4.5 | 1.0 |
| $\beta_{\text{KL}}$ | | | | | | | | 0.04 | 0.01 | |
| Learning rate | 1e-4 | **1e-4** | 1e-4 | **1e-4** | **1e-5** | **3e-4** | 1e-5 | 3e-4 | 3e-4 | 1e-5 |
| Max grad norm | 1.0 | **1.0** | 1.0 | **1.0** | **1.0** | **1.0** | 1.0 | 1.0 | 1.0 | 1.0 |
| Prompts per iteration | 256 | **512** | 288 | **576** | **32** | **32** | 32 | 8 | 32 | 32 |
| Images per prompt | 2 | **2** | 2 | **2** | **16** | **12** | 16 | 8 | 16 | 12 |
| Images per iteration | | | | | **512** | **384** | 512 | 64 | 512 | 384 |
| Gradient updates | | | | | **2** | **4** | 2 | 2 | 2 | 4 |
| Clip range ($\xi$) | 1e-4 | **1e-4** | 1e-4 | **1e-4** | **1e-4** | **1e-4** | 1e-4 | 1e-4 | 1e-4 | 1e-4 |
| LoRA ($r, \alpha$) | (4,4) | **(4,4)** | (4,4) | **(4,4)** | | **(128,256)** | | (32,64) | (32,64) | |
| Checkpoint freq. | 1 | **1** | 1 | **1** | **50** | **60** | 50 | 20 | 20 | 60 |

**Prompts/iteration & Images/prompt.** These specify the number of unique prompts sampled per training iteration, and the number of images generated per prompt. For DDPO, `Images per iteration` random prompts are sampled each iteration, as it does not require group-level normalization.

**LoRA.** $r$ and $\alpha$ are the rank and scaling factor for LoRA fine-tuning, respectively. An empty entry indicates full fine-tuning.

## B.2 EXPERIMENT-SPECIFIC RATIONALE & LIMITATIONS

### B.2.1 DDPO

**Rationale:** This experiment replicates the original DDPO setup to establish a direct baseline. The goal is to evaluate PCPO's impact on stability and sample fidelity within a simple, well-understood environment. The lightweight nature of the task allows for checkpointing every epoch, enabling us to pinpoint the exact epoch where reward levels converge on a validation set of 50K seed-controlled images. This facilitates a rigorous statistical analysis using LMMs to isolate PCPO's effect from prompt-level variance.

**Limitations:** Crucially, this setup is intentionally limited to simple prompts (45 animal nouns for Aesthetics, 45 animal nouns $\times$ 3 activity verbs for BERTScore). As noted in Wallace et al. (2024), the DDPO framework does not generalize well to complex, open-vocabulary prompts. Therefore, this experiment evaluates performance on a narrow task, not general prompt-following ability. Open-vocabulary experiments based on DanceGRPO and Flow-GRPO are presented later in the paper to address this.

**Miscellaneous Details:** For BERTScore, we replace the LLaVA VLM used in the original implementation with Qwen2.5-VL-3B-Instruct (Bai et al., 2025).

### B.2.2 DANCEGRPO

**Rationale:** This experiment evaluates PCPO in a more complex, open-vocabulary setting, using the DanceGRPO framework (Xue et al., 2025) and the HPDv2 prompt set. This setup tests PCPO's ability to maintain fidelity and diversity while optimizing for high-level human preferences (HPS).

**Limitations:** The DanceGRPO framework is computationally intensive, limiting the number of ablations and repetitions we can perform. Most ablations are conducted on the simpler DDPO setup. For FLUX, we performed LoRA fine-tuning on 8 GPUs, using the official scripts provided by the authors. This differs from the 16-GPU full fine-tuning in the original paper, and results in a slightly lower performance ceiling. DanceGRPO's configurations for FLUX performs 4 gradient updates per iteration and uses timestep subsampling, making the training dynamic more fragile.

**Miscellaneous Details:** For SD, we mainly use SD1.4 following Xue et al. (2025). For additional experiments, we use SD1.5 as the base model to align with other DPO-based frameworks. Training was terminated at 240 epochs, as further training degraded image quality for the baseline with minimal reward gains (Figure 20). Note that the naive acceleration ablation (Figure 7(d)) was run using full fine-tuning on 8 H100 GPUs.

### B.2.3 FLOW-GRPO

**Rationale:** To demonstrate maximum generalizability, this experiment applies PCPO to a completely different regime: a different model (SD3.5-M), noise schedule (Flow-GRPO), rewards (OCR, PickScore), and training objective (has a KL penalty). Success here shows that PCPO is a model-agnostic principle, not a quirk of one specific setup.

**Limitations:** Due to resource constraints, we conduct experiments on the resource-light `pickscore_sd3_4gpu` and `ocr_sd3_1gpu` setups from the Flow-GRPO codebase, not the full-scale setups in the original paper.

**Miscellaneous Details:** To run code on GPUs with 24GB VRAM, we modified the prompt sampler. In this process, we improved upon the original implementation by guaranteeing the number of unique prompts per iteration match the configured value. For PickScore experiments, we used the Pick-a-Pic

SFW sub-dataset instead of the Pick-a-Pic dataset used in the original paper, to avoid generating NSFW content. KL divergence was calculated as:

$$D_{\mathrm{KL}}(\pi_\theta||\pi_{\mathrm{ref}}) = \frac{\Delta t}{2}\left(\frac{\sigma_t(1-t)}{2t} + \frac{1}{\sigma_t}\right)^2\|\mathbf{u}_\theta(\mathbf{x}_t, t) - \mathbf{u}_{\mathrm{ref}}(\mathbf{x}_t, t)\|^2$$

following Liu et al. (2025), which avoids `exp()` operations and thus maintains numerical stability (see Appendix D).

### B.3 EVALUATION DETAILS

#### B.3.1 IMAGE QUALITY METRICS

For a controlled evaluation on image quality metrics, we generate a fixed set of 50,000 images (45 prompts $\times$ 1112 seeds). This set is produced by the original base model to serve as a reference, and by each fine-tuned policy for comparison. We evaluate these images at two levels of granularity:

- **Overall Metrics:** We compute FID, $FD_{\mathrm{DINO}}$ and IS across the entire 50k generated samples to measure overall fidelity and quality.
- **Per-Prompt Analysis:** For a more fine-grained statistical analysis, we compute metrics on a per-prompt basis (i.e., on the 1112 images generated for each prompt). We use a LMM on these per-prompt scores to isolate our method's true impact from prompt-level variance. Coefficients were estimated via Restricted Maximum Likelihood (REML). At this level, we also compute LPIPS Diversity (Zhang et al., 2018) to measure intra-prompt variety.

It is important to note that FID and $FD_{\mathrm{DINO}}$ scores computed on smaller sample sets (e.g., FD per 1.1k images) are expected to have a higher magnitude than scores computed on the full dataset (FD-50k). Consequently, the effect size ($\beta_{alg}$) estimated by the LMM, which is based on these per-prompt scores, may appear larger in magnitude than the simple difference observed in the overall FD-50k results. Since FD scores are calculated against the outputs of the original base model, they can be interpreted as a measure of distributional drift; a lower score signifies less deviation from the original policy. This indicates a successful mitigation of mode collapse and serves as a strong proxy for the preservation of the base model's inherent fidelity and diversity.

#### B.3.2 ALIGNMENT & GENERALIZATION METRICS

In addition to quality metrics, we evaluate on a suite of preference alignment metrics (HPSv2.1, CLIP-Score, PickScore, ImageReward, Aesthetics). These are used in Table 5 to measure generalization to unseen prompts and reward models. The MJHQ-30K dataset contained 3.3K prompts that do not fit in the CLIP tokenizer ($> 77$ tokens). As such, we filtered these out and used the first 5K prompts of the remaining set for evaluation. For Table 9, we use HPSv2 instead of HPSv2.1 to align with prior works. We used the 2048 prompts from the Pick-a-Pic evaluation set provided by the Flow-GRPO codebase. For all prompt datasets, we generated 1 seed-controlled image per prompt.

#### B.3.3 HUMAN PREFERENCE STUDY

To ensure a fair comparison that accounts for PCPO's accelerated convergence, we conducted a "bracketing" human preference study. Our model (epoch 120) was compared against two baseline DanceGRPO checkpoints that bracket its reward level: epoch 180 (lower reward) and epoch 240 (higher reward). The evaluation used a pool of 450 unique image pairs for each comparison (45 prompts $\times$ 10 seeds). To eliminate bias, a web interface presented these pairs by fully randomizing the comparison set, prompt, specific image, and left/right display position for each question. We collected 297 responses against epoch 180 and 332 responses against epoch 240.

#### B.3.4 PLOTTING AND VISUALIZATION DETAILS

All training curves and clipping fractions, unless specified otherwise, are smoothed with a moving average window of 5 epochs to improve readability.

**Figure 5.** Subfigures (a, b) plot the validation reward, unlike other plots that show the reward computed on training samples each iteration. This leads to clearer trends, as the training reward is

Table 8: Computational resources used for each experimental setup.

| Experiment | Hardware |
|---|---|
| DDPO (Aesthetics) | 2x NVIDIA RTX 4090 (24GB) |
| DDPO (BERTScore) | 3x NVIDIA RTX 4090 + 1x VLM GPU |
| DDPO (Ablations) | 2x NVIDIA RTX 3090 (24GB) |
| DanceGRPO (SD1.4) | 8x NVIDIA A100 (40GB) |
| DanceGRPO (SD1.5) | 8x NVIDIA A40 (48GB) |
| DanceGRPO (FLUX) | 8x NVIDIA A5000 (24GB) |
| DanceGRPO (Naive Acc.) | 8x NVIDIA H100 (80GB) |
| Flow-GRPO (OCR) | 1x NVIDIA RTX3090 |
| Flow-GRPO (PickScore) | 4x NVIDIA RTX3090 |

noisy due to randomness from prompts sampled at each iteration. Validation rewards are computed every 50 epochs on a fixed set of 1024 prompts that are not in the training pool.

Subfigures (c, d) plot the Pareto frontier between reward and KL divergence with respect to the reference policy (base model; not fine-tuned). To reduce noise in the visualizations, the KL divergence is plotted using a rolling mean of all estimated values up to the current epoch. While this approach yields a low-variance estimate, it likely underestimates the true KL divergence at that specific epoch. Both the baseline and PCPO start at the top-left corner (low reward, low KL divergence), and move towards the bottom-right corner (high reward, high KL divergence) as training progresses.

### B.4 COMPUTATIONAL RESOURCES

RL-based fine-tuning is known to be sensitive to hardware and implementation details. We report our resources for full reproducibility in Table 8.

## C COMPARISON WITH DPO AND SELF-PLAY BASELINES

For completeness, we attempt to provide comparisons with reward-free methods. While this attempt is meaningful, we emphasize that direct, fair comparison between reward-based methods (such as PCPO) and reward-free methods presents significant methodological difficulties that render a head-to-head evaluation inconclusive.

**Difficulty 1: Establishing a Fair Evaluation Protocol.** The core challenge lies in the choice of evaluation metric and the fairness of access to human preference data.

- **Restricting Reward Model Choice:** Limiting reward-based methods to a single model (e.g., PickScore, trained on a subset of Pick-a-Pic prompts) creates an unfair disadvantage. Reward-based methods inherently rely on "distilled" knowledge from human annotations, whereas reward-free methods learn directly from preference data. Moreover, it restricts reward-based methods from leveraging their ability to aggregate preference knowledge from multiple sources (e.g., linear combinations of reward models, as explored in Xue et al. (2025); Zheng et al. (2025)).

- **Unrestricted Reward Model Choice:** Allowing reward-based methods to use a linear combination of multiple reward models or custom reward functions is then argued to be unfair to reward-free methods, which must "zero-shot" against a richer evaluation metric they were not trained to optimize. It also allows reward-based methods to indirectly access a wider set of human preference data distilled into various reward models.

Similar to prior work comparing DPO methods against policy-gradient methods (Liang et al., 2025; Zhang et al., 2025b), we set our experimental protocol to place PCPO at a comparative disadvantage by using a single, fixed reward model (PickScore).

**Difficulty 2: Reproducibility and Closed-Weight Models.** A second major hurdle is the lack of open-source checkpoints for SOTA DPO or self-play baselines, including SSPO (Zhang et al., 2025a),

Table 9: Comparison with DPO and Self-Play baselines on SD1.5, evaluated on preference metrics from the Pick-a-Pic dataset. Scores of other works are cited from each paper. †: trained on Pick-a-Pic v2. More details in text.

| Method | HPSv2 (↑) | Aesthetic (↑) | PickScore (↑) | ImgRwd (↑) |
|---|---|---|---|---|
| SD1.5 (Base) | 0.261 | 5.43 | 20.09 | 0.09 |
| PCPO (E200) | 0.271 | 6.14 | 22.42 | 0.99 |
| gains (E200) | +0.010 | **+0.71** | **+2.33** | **+0.90** |
| SD1.5 (Base) | 0.238 | 5.37 | 20.53 | -0.16 |
| SSPO † | 0.272 | 5.94 | 21.90 | 0.70 |
| gains | **+0.034** | +0.57 | +1.37 | +0.86 |
| SD1.5 (Base) | 0.271 | 5.77 | 21.18 | 0.92 |
| SPIN-Diffusion_Iter3 | 0.276 | 6.25 | 22.00 | 1.12 |
| gains | +0.005 | +0.48 | +0.82 | +0.20 |
| SD1.5 (Base) | 0.265 | 5.47 | 20.56 | 0.08 |
| LPO | 0.276 | 5.95 | 21.69 | 0.66 |
| gains | +0.011 | +0.48 | +1.13 | +0.58 |

SPIN-Diffusion (Yuan et al., 2024), and RainbowPA (Sun et al., 2025). Furthermore, while most work report similar evaluation protocols, i.e. slicing the Pick-a-Pic dataset into training and validation sets, the exact details of implementation differ significantly across published works. Notably, the reported base model (SD 1.5) performance varies widely, suggesting substantial variation in the difficulty and composition of the validation prompts used. Moreover, some works (Zhang et al., 2025a) use the richer Pick-a-Pic v2 dataset, while others apply filtering (Zhang et al., 2025b).

A head-to-head comparison is thus difficult, and reported results must be interpreted with caution. While we attempt to normalize for prompt differences by comparing relative gains (final score minus reported base score), this metric can still favor models that report an atypically low base score. For instance, SSPO reports a significant gain of 0.034 in HPSv2, but its final performance is commensurate with that of other methods, suggesting their validation set was uniquely challenging for the base model, inflating the relative gain metric.

**Comparison Results.** Nonetheless, we present a quantitative comparison between PCPO and leading DPO/self-play baselines in Table 9. PCPO was trained using the standard DanceGRPO setup on SD1.5, using Pick-a-Pic prompts and the PickScore reward model. PCPO demonstrates highly competitive performance, underscoring its efficacy in human preference alignment. While CLIPScore comparisons are omitted due to their absence in prior literature, we note that PCPO preserves semantic fidelity, maintaining a score of 0.387 (identical to the base model).

# D  ADDITIONAL RESULTS

**Speedups for Different Reward Thresholds.** Table 10 details the training efficiency gains of PCPO across all experimental settings. Speedup is calculated based on the number of epochs required to reach specific reward thresholds during training. As training involves stochastic sampling of prompts and seeds, speedup values are subject to noise from the stochastic sampling of prompts and random seeds per iteration.

**More Convergence Plots.** Figure 8 shows the reward optimization traces and clipping fractions for the default, smaller batch size of 256 in DDPO.

**Wall-Clock Time and Computational Cost.** PCPO introduces no computational overhead compared to the baseline (Figure 9). While baselines require an additional backward SDE sampler step to compute $p_\theta$ for policy updates, PCPO only requires the network forward pass for noise/velocity prediction, which baselines also compute. By saving this SDE sampler step, PCPO achieves a slightly lower wall-clock time per epoch.

**Soundness of** $\log \rho_t \approx \rho_t - 1$ **Approximation.** We justify our choice of using the log-ratio $\log \rho_t$ in the PCPO objective (Eq. (3)) via a Taylor approximation from four aspects.

Table 10: Training efficiency comparison across all experimental settings. We report the number of epochs for the baseline and our method (PCPO) to reach various target reward levels. Speedup is calculated as $(\text{Epochs}_{\text{Baseline}}/\text{Epochs}_{\text{PCPO}} - 1) \times 100\%$. Highlighted rows correspond to the primary reward targets discussed in the main text.

| Framework | Setting | Reward | Baseline | PCPO | Speedup |
|---|---|---|---|---|---|
| DDPO (SD1.5) | Aesthetics (256/128) | **6.90** | **260** | **131** | **98.5%** |
| | | 6.60 | 210 | 100 | 110.0% |
| | | 6.30 | 165 | 70 | 135.7% |
| | | 6.00 | 108 | 48 | 125.0% |
| | Aesthetics (512/256) | **6.90** | **147** | **118** | **24.6%** |
| | | 6.60 | 107 | 89 | 20.2% |
| | | 6.30 | 73 | 61 | 19.7% |
| | | 6.00 | 47 | 44 | 6.8% |
| | BERTScore (288/144) | **0.52** | **246** | **198** | **24.2%** |
| | | 0.51 | 188 | 175 | 7.4% |
| | | 0.50 | 145 | 155 | -9.4% |
| | | 0.49 | 90 | 112 | -19.7% |
| | BERTScore (576/288) | **0.52** | **191** | **146** | **30.8%** |
| | | 0.51 | 159 | 124 | 28.2% |
| | | 0.50 | 125 | 89 | 40.4% |
| | | 0.49 | 86 | 74 | 16.2% |
| DanceGRPO (SD1.4) | HPSv2.1 | **0.37** | **236** | **188** | **25.5%** |
| | | 0.36 | 173 | 153 | 13.1% |
| | | 0.35 | 153 | 124 | 23.4% |
| | | 0.34 | 116 | 102 | 13.7% |
| DanceGRPO (FLUX) | HPSv2.1 | **0.36** | **209** | **148** | **41.2%** |
| | | 0.35 | 148 | 93 | 59.1% |
| | | 0.34 | 127 | 83 | 53.0% |
| | | 0.33 | 101 | 50 | 102.0% |

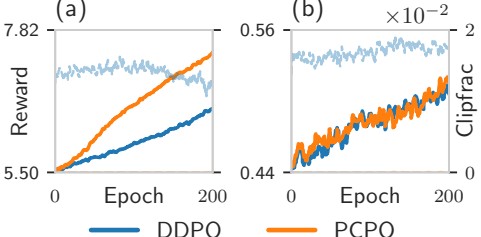

Figure 8: Training trajectories on smaller batch for PCPO (orange) and DDPO (blue) on (a) Aesthetics, (b) BERTScore rewards.

Figure 9: Wall-clock time per epoch for DanceGRPO (blue) and PCPO (orange) for (a) SD1.4, (b) FLUX training.

1. **Empirical Distribution of $\rho_t$:** We tracked the distribution of $\rho_t$ during training (from scratch and near convergence) for both diffusion (DDPO, SD1.5) and flow models (DanceGRPO, FLUX). Figure 10 shows that $\rho_t$ remains close to 1.0, with a standard deviation on the order of $10^{-4}$. This confirms that the policy operates precisely in the region where the approximation is valid, making the Taylor approximation error negligible.

2. **Error Bounding:** The clipping mechanism of PPO and GRPO provides a strong bound on the approximation error. In T2I alignment, the clip range $\xi$ is typically set to a very small value (e.g., $10^{-4}$ for all of our experiments). The gradient in our objective (Eq. (3)) only flows when $|\log \rho_t|$ is within this clipping range. This means the Taylor approximation is

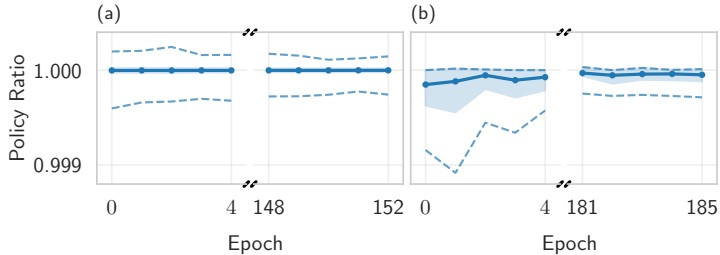

Figure 10: Empirical distribution of the policy ratio $\rho_t$ during training for (a) DanceGRPO (FLUX) and (b) DDPO (SD1.5). In all cases, the mean, min, and max values remain exceptionally close to 1.0, validating the Taylor approximation $\log \rho_t \approx \rho_t - 1$. Shade heights correspond to $2 \times \text{std}(\rho_t)$.

---

**Algorithm 1** Ratio Computation

| **Baseline (PPO/GRPO)** | **PCPO (Ours)** |
|---|---|
| 1: $\log p_\theta \leftarrow \text{policy.eval(state, action)}$ | 1: $\log p_\theta \leftarrow \text{policy.eval(state, action)}$ |
| 2: $\log p_{\text{old}} \leftarrow \text{old\_policy.eval(state, action)}$ | 2: $\log p_{\text{old}} \leftarrow \text{old\_policy.eval(state, action)}$ |
| 3: $\log \rho \leftarrow \log p_\theta - \log p_{\text{old}}$ | 3: $\log \rho \leftarrow \log p_\theta - \log p_{\text{old}}$ |
| 4: $\rho \leftarrow \exp(\log \rho)$  *// Unstable step* | 4:  *// No exp() step* |
| 5: $\text{loss} \leftarrow f(\rho, A)$ | 5: $\text{loss} \leftarrow f(\log \rho, A)$  *// Stable* |

---

only ever utilized in the exact region where it is most accurate ($|\rho_t - 1| \approx |\log \rho_t| < \xi \ll 1$), making the effective approximation error negligible (on the order of $\mathcal{O}(\xi^2)$).

3. **Numerical Stability:** From a numerical perspective, we argue that a more significant source of instability—which our $\log \rho_t$ formulation *also* fixes—is the numerical precision error in computing $\rho_t$ itself. As shown in Algorithm 1, standard PPO/GRPO must compute $\rho_t = \exp(\log \pi_\theta - \log \pi_{\text{old}})$. PCPO, in contrast, operates *directly* in log-space by using $\log \rho_t = \log \pi_\theta - \log \pi_{\text{old}}$ in the objective. While $\rho_t$ should be strictly 1.0 for an on-policy update, our per-step measurements typically show a mean value of 0.99997 to 0.99998. This suggests that the floating point error is larger than the Taylor expansion error $\sim \xi^2/2$.

4. **Theoretical Justification:** Lastly, even if the approximation were to fail, our formulation remains sound. As shown by Huang et al. (2024), the objective from Eq. (2) is a hinge loss where $\rho_t - 1$ acts as a "classifier." Their work demonstrated that this term can be swapped with $\log \rho_t$ (among other variants $\sqrt{\rho_t} - 1$, $\pi_\theta - \pi_{\text{old}}$) while yielding similar performance. This validates our log-hinge objective as a robust choice on its own merits, independent of the approximation.

**Gradient Stability Analysis.** To provide a quantitative analysis of training stability, we measured the mean of the absolute gradient, $\mathbb{E}[|g_t|]$, and the variance of the absolute gradient, $\text{Var}(|g_t|)$, for the first layer of the U-Net or transformer. This analysis was performed on both DDPO (SD1.5) and DanceGRPO (FLUX) frameworks during two training phases: (i) the initial 5 epochs (starting from scratch) and (ii) 5 epochs near convergence (loaded from checkpoints at target reward levels).

The results, shown in Figure 11, confirm PCPO's superior stability. While the gradient statistics are similar for both methods at the very start of training, a significant gap emerges as training progresses. Near convergence, the baseline models exhibit gradients with larger mean and variance. This provides direct, quantitative evidence that PCPO mitigates gradient instabilities, leading to smoother and more stable training dynamics.

# E   CONNECTION TO REINFORCE ELIGIBILITY VECTOR

To formalize the connection between our proportionality principle and established RL theory, we analyze the REINFORCE algorithm (Sutton & Barto, 2018; Williams, 1992). REINFORCE provides a first-principles, Monte Carlo estimator for the policy gradient.

For an episodic, non-discounted ($\gamma = 1$) task with a terminal reward, the parameter update is a sum over all timesteps $t$. In our setting, the return $G_t$ is the same terminal advantage $A$ for all steps. The

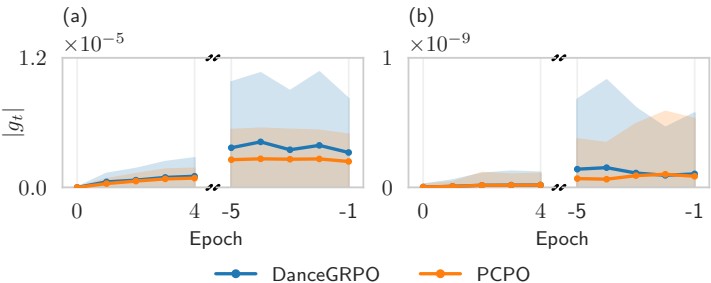

Figure 11: Plots of $|g_t|$ for the network's first layer during training. Shades indicate $2 \times \text{std}(|g_t|)$.

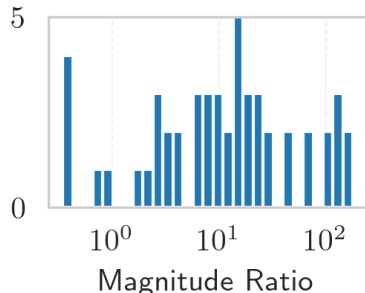

Figure 12: Magnitude ratio of the dot product term to the squared norm term in Equation 6 for the first off-policy data point in DDPO. The dot product term is dominant even in the off-policy case, justifying our simplification.

update rule is therefore:

$$\Delta\theta \propto \sum_{t=1}^{T} A\mathbf{e}_t(\theta), \quad \mathbf{e}_t(\theta) = \nabla_\theta \log \pi_\theta(a_t|s_t) \tag{15}$$

As noted by Sutton & Barto (2018), the term $\mathbf{e}_t(\theta)$ is the *eligibility vector*. This vector is the core component containing the policy's parameter gradients, and is the "only place that the policy parameterization appears in the algorithm." Assuming each action contributes equally to the final reward, REINFORCE assigns credit scales each eligibility vector by the same advantage $A$ for fair credit assignment.

We can now draw a direct analogy to our on-policy case (justified in Figure 12). From our analysis, the gradient contribution from a single timestep $t$ is proportional to:

$$\Delta\theta_t \propto A \cdot w(t) \cdot \underbrace{\left[(\nabla_\theta \hat{\boldsymbol{\varepsilon}}_\theta^{(t)} \cdot \boldsymbol{\epsilon}_{\text{old}}^{(t)})\right]}_{\text{Analogous to } \mathbf{e}_t(\theta)} \tag{16}$$

The term $\nabla_\theta \hat{\boldsymbol{\varepsilon}}_\theta^{(t)} \cdot \boldsymbol{\epsilon}_{\text{old}}^{(t)}$ is our core policy gradient, which is analogous to the eligibility vector $\mathbf{e}_t(\theta)$. However, as Equation 16 shows, this "eligibility vector" is multiplied by $w(t)$ *before* being scaled by the advantage $A$. Assuming the expected statistics of the dot product term are constant across timesteps, the overall gradient magnitude for each step becomes directly proportional to its weight, $w(t)$.

This $w(t)$ is an arbitrary scaling factor that arises from the sampler's mathematics, not a deliberate credit assignment choice; i.e. proportional to the integration interval $\Delta t$. It breaks the uniform credit assignment of the REINFORCE framework by non-uniformly and inconsistently scaling the gradient contribution from each step. PCPO restores sound credit assignment by ensuring that $w(t)$ is proportional to the timestep $\Delta t$, its true contribution to the trajectory.

# F    IMPLICIT REWARD GUIDANCE

We introduce *Implicit Reward Guidance (IRG)*, an inference-time scaling mechanism that mitigates reward overoptimization by interpolating between a base model and one or more RL fine-tuned models. This principle was concurrently developed in the Reinforcement Learning Guidance (RLG) framework (Jin et al., 2025); our work further extends the concept by demonstrating its application to dynamically compose multiple learned rewards.

The theoretical motivation for this stems from D3PO (Yang et al., 2024), which interpreted the learned policy as implicitly encoding a reward signal $\log\left(p_\theta(\mathbf{x}_0 \mid \mathbf{x}_t)/p_{\text{ref}}(\mathbf{x}_0 \mid \mathbf{x}_t)\right)$. Following CFG++ (Chung et al., 2025), IRG frames the sampling process as an optimization problem where, given a noisy state $\mathbf{x}_t$, we aim to find a sample $\mathbf{x}_0$ that maximizes the implicit reward function.

To guide the sampling process, we can take a gradient step in the direction that maximizes this reward. Approximating the predicted denoised sample $\hat{\mathbf{x}}_0$ as a projection of $\mathbf{x}_t$, the gradient of the reward with respect to $\hat{\mathbf{x}}_0$ simplifies to:

$$\nabla_{\hat{\mathbf{x}}_0} \log \frac{p_\theta(\mathbf{x}_0 \mid \mathbf{x}_t)}{p_{\text{ref}}(\mathbf{x}_0 \mid \mathbf{x}_t)} \approx \frac{1}{\sigma_t^2}(\boldsymbol{\mu}_\theta(\mathbf{x}_t) - \boldsymbol{\mu}_{\text{ref}}(\mathbf{x}_t)),$$

where $\boldsymbol{\mu}_\theta$ and $\boldsymbol{\mu}_{\text{ref}}$ are the means of the one-step reverse transition distributions for the target and reference policies, respectively. This gradient term represents the direction of maximal reward increase.

While CFG++ adds this guidance term of the denoised sample $\hat{\mathbf{x}}_0$ before renoising, this can be unstable for IRG. In IRG, the guidance term comes from a separate model that is distinct from the generative model being used for sampling. Using different models to compute separate noise vectors for the Tweedie mean estimation and the renoising step can destabilize the DDIM inversion process.

Therefore, we propose an anlogous but more stable formulation where the guidance is applied directly to the noise prediction. By defining a guided noise estimate $\hat{\boldsymbol{\varepsilon}}^{\text{IRG}}(\mathbf{x}_t) := \hat{\boldsymbol{\varepsilon}}_{\text{ref}}(\mathbf{x}_t) + \lambda(\hat{\boldsymbol{\varepsilon}}_\theta(\mathbf{x}_t) - \hat{\boldsymbol{\varepsilon}}_{\text{ref}}(\mathbf{x}_t))$, we use this single, consistent noise vector for both the Tweedie mean estimation and the renoising step of the DDIM update. This ensures stability while effectively steering the generation process toward higher-reward outcomes.

Furthermore, IRG can compose multiple learned rewards. Given $n$ models $\{\theta_i\}_{i=1}^n$, each aligned to a distinct reward, IRG forms a mixed predictor:

$$\hat{\boldsymbol{\varepsilon}}_{\{\theta_i\}}^{\text{IRG}} = \hat{\boldsymbol{\varepsilon}}_{\text{ref}} + \sum_{i=1}^n \lambda_i\left(\hat{\boldsymbol{\varepsilon}}_{\theta_i} - \hat{\boldsymbol{\varepsilon}}_{\text{ref}}\right).$$

This principle extends directly to flow models by replacing the noise predictor $\hat{\boldsymbol{\varepsilon}}$ with the velocity predictor $\hat{\mathbf{u}}$:

$$\hat{\mathbf{u}}_{\{\theta_i\}}^{\text{IRG}} = \hat{\mathbf{u}}_{\text{ref}} + \sum_{i=1}^n \lambda_i\left(\hat{\mathbf{u}}_{\theta_i} - \hat{\mathbf{u}}_{\text{ref}}\right).$$

As shown in Figure 13, IRG effectively mitigates reward overoptimization by tuning the guidance scale $\lambda$ at inference time. It also enables the flexible composition of multiple objectives, providing a smooth interpolation between competing rewards like aesthetics and text-alignment (Figure 14).

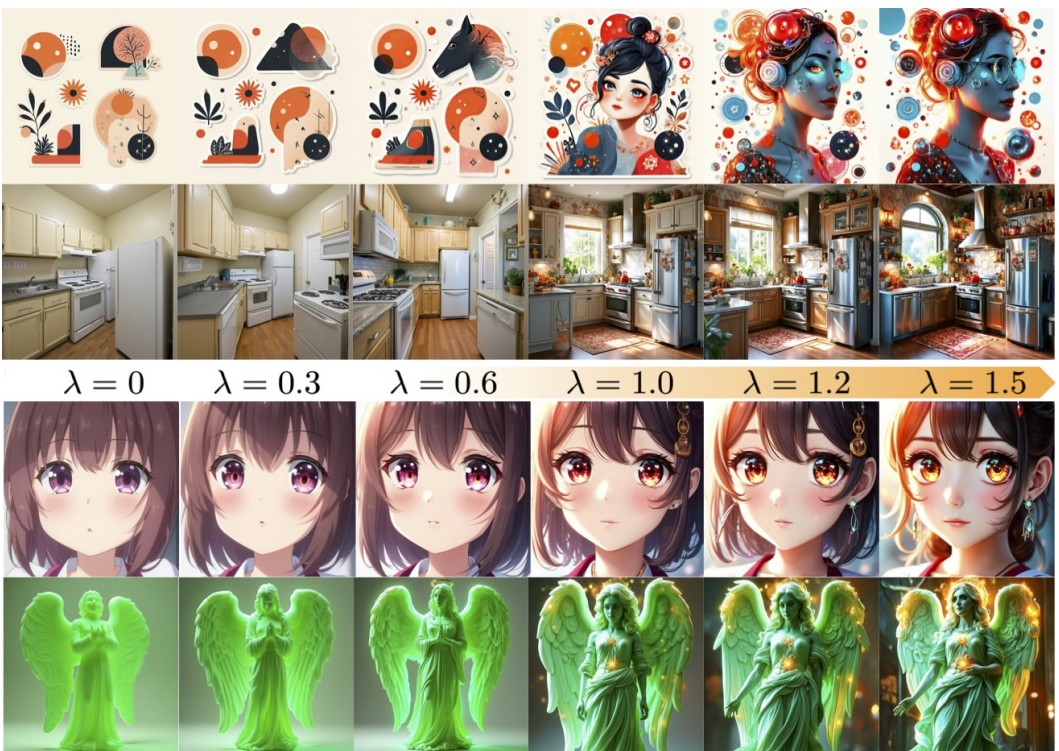

Figure 13: **Test-time scaling via IRG at various $\lambda$ values:** Top rows: Reducing the weight to $\lambda < 1$ interpolates between the base model and fine-tuned model, alleviating reward overoptimization at inference-time. Bottom rows: Increasing the weight to $\lambda > 1$ extrapolates the internal reward, thus can be used to enhance visual appeal. Full prompt list in Appendix G.

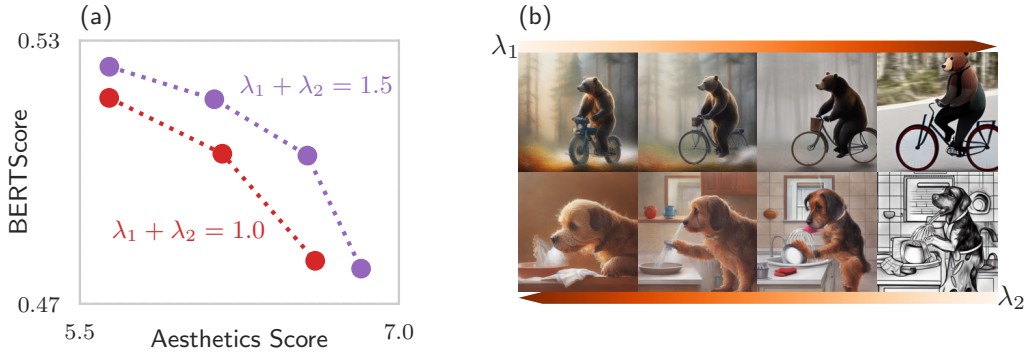

Figure 14: IRG tradeoff. (a) Aesthetics-BERTScore curves for different guidance scales, showing that increasing aesthetic guidance $\lambda_1$ boosts visual quality but reduces semantic alignment, and vice versa for semantic guidance $\lambda_2$. (b) Generated samples for two prompts (top: a bear riding a bike; bottom: a dog washing the dishes) arranged by increasing $\lambda_1$ (left arrow) and $\lambda_2$ (right arrow) for $\lambda_1 + \lambda_2 = 1.5$, demonstrating smooth interpolation between aesthetic and semantic objectives.

## G    FULL PROMPTS FOR FIGURES

**Figure 1:**   (a) **Aesthetics reward:** *cat, bear, butterfly, chicken, bee, camel.*

(b) **HPSv2.1 reward:**

* *A Thangka painting depicting the devil controlling a group of people in a circular formation.*
* *A painting of a monkey wearing gold headphones and sunglasses looking up at a starry night sky.*
* *Matador challenging a bull in a desert-filled ramen bowl.*

**Figure 13:**

– *Minimalist sticker art featuring abstract designs by Victor Ngai, Kilian Eng, and Lois Van Baarle.*
– *The kitchen has a stove and a refrigerator.*
– *A close-up portrait of a cute anime girl with extremely detailed eyes, featured as a key visual in official media.*
– *A green 3D-printed Biblically accurate angel.*

**Figure 17**

– *a bat playing chess*
– *a bird riding a bike*
– *fish*
– *lion*

**Figure 18**

– *A portrait of a lion goddess wreathed in flame, posing in full body.*
– *A portrait painting of Stephany Eisnor.*
– *The kitchen has a stove and a refrigerator.*

**Figure 19**

– *Several people waiting at a bus stop in a dark city night, depicted in a digital illustration.*
– *Side-view portrait of a knight with a skull helmet adorned with spikes, depicted in a tenth century stained glass window.*
– *The album cover for the band Underealm features an evil entity in a sophisticated suit, with dark and intricate details.*

**Figure 20**

– *A Thangka painting depicting the devil controlling a group of people in a circular formation.*
– *A painting of a monkey wearing gold headphones and sunglasses looking up at a starry night sky.*
– *Matador challenging a bull in a desert-filled ramen bowl.*
– *Several people waiting at a bus stop in a dark city night, depicted in a digital illustration.*
– *Side-view portrait of a knight with a skull helmet adorned with spikes, depicted in a tenth century stained glass window.*
– *The album cover for the band Underealm features an evil entity in a sophisticated suit, with dark and intricate details.*
– *The image features a copper material with references to various art platforms and artists.*
– *Anor Londo, Dark Souls, with a dragon flying in the distance at night.*
– *Two Tyrannosaurus rexes engaged in a boxing match.*
– *The image depicts Yoshi Island created by Beeple.*
– *Digital painting featuring Soviet realism and grunge elements with a range of artistic influences, created by multiple artists and showcased on ArtStation.*
– *An artwork from Dan Mumford collection featuring a mage invoking divine gods during a storm with lightnings.*

# H    MORE QUALITATIVE RESULTS

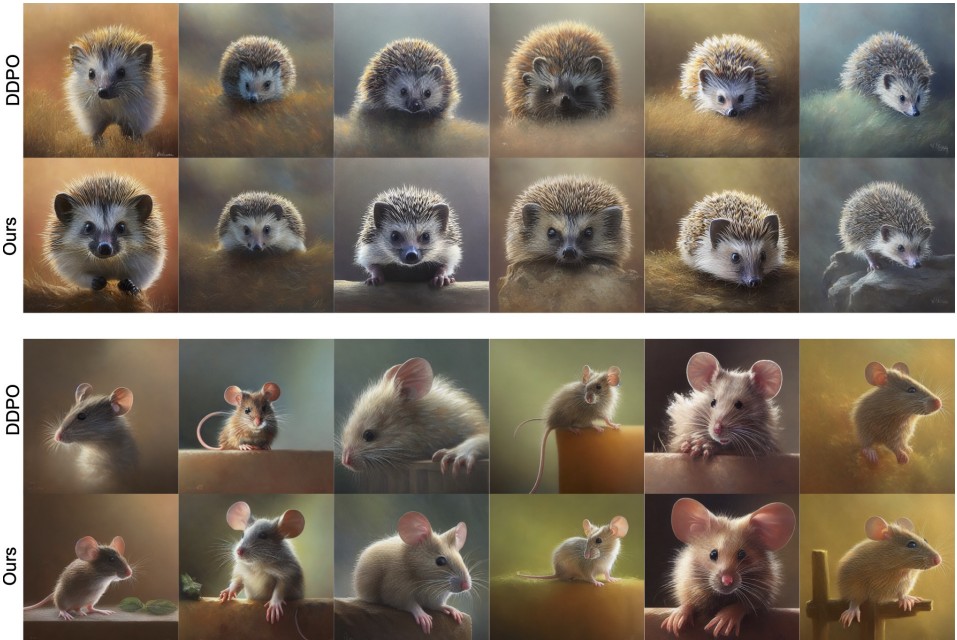

Figure 15: **PCPO mitigates mode collapse at high reward levels.** For the prompts "hedgehog" and "mouse" (6 consecutive seeds), DDPO (top) exhibits noticeable mode collapse. In contrast, PCPO (bottom) maintains high visual diversity and sharpness at the same reward level.

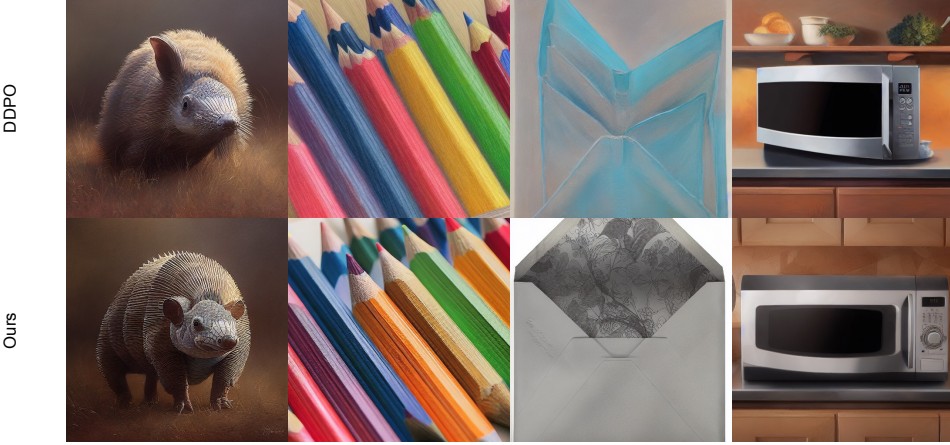

Figure 16: **PCPO generalizes better to unseen prompts than DDPO.** For prompts that were not in the fine-tuning prompt dataset ("armadillo", "colored pencils", "envelope", "microwave"), PCPO retains the generation capabilities much better than the baseline DDPO.

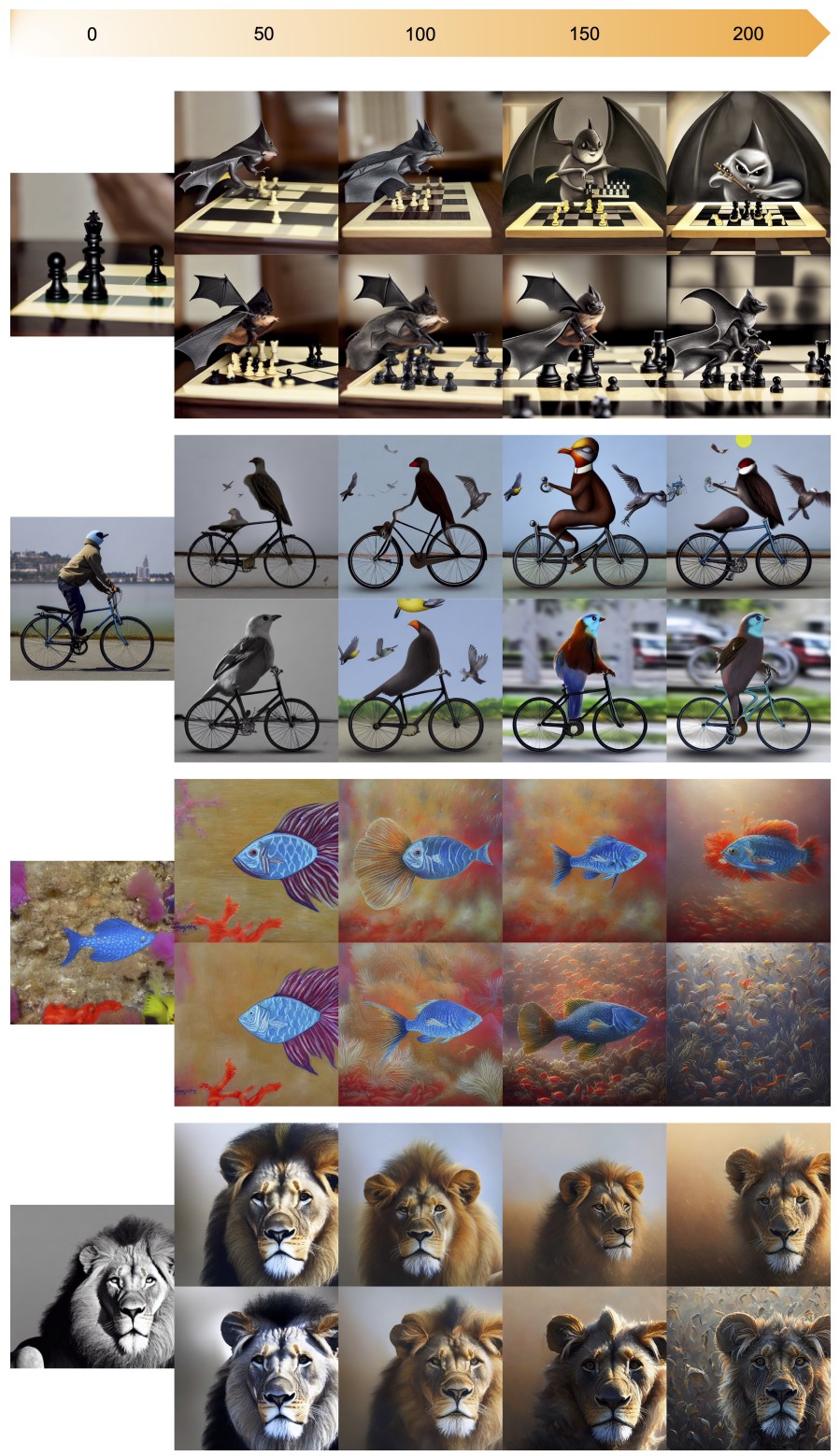

Figure 17: **PCPO demonstrates superior training stability, avoiding premature collapse.** This figure tracks the qualitative progression per epoch for DDPO (top rows) and PCPO (bottom rows). The baseline DDPO shows early image degradation across both BERTScore (first two groups) and Aesthetics (last two groups) rewards. PCPO consistently maintains higher fidelity, although it can also exhibit collapse at extremely high reward levels (e.g., Aesthetics at epoch 200).

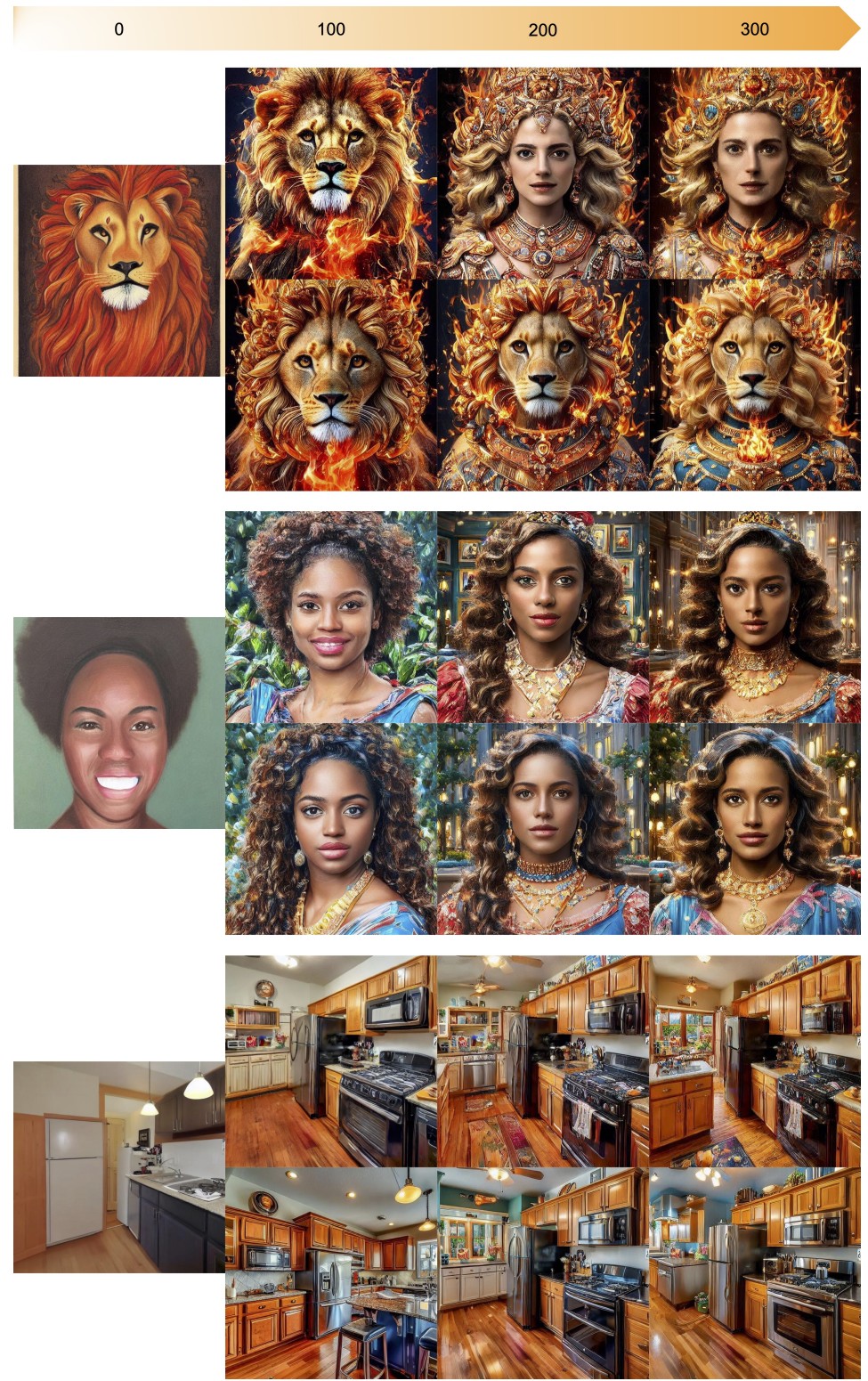

Figure 18: **DanceGRPO SD1.4, progression per epoch.** Within each group, the top row shows DanceGRPO (SD1.4) results, the bottom row shows PCPO. PCPO better preserves fidelity and text-image alignment, DanceGRPO exhibits noticeable model collapse.

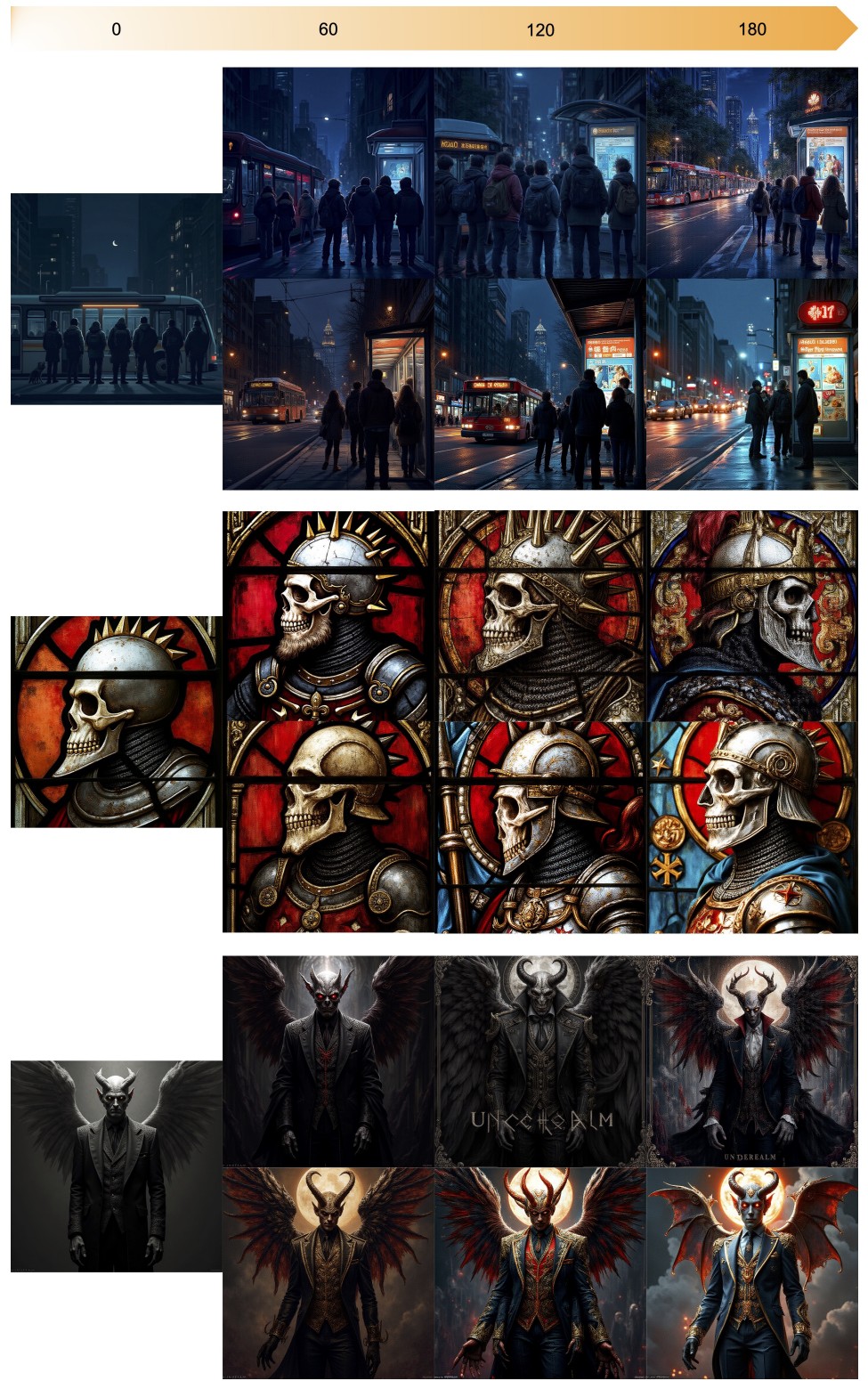

Figure 19: **DanceGRPO FLUX, progression per epoch.** Within each group, the top row shows DanceGRPO (SD1.4) results, the bottom row shows PCPO. PCPO better preserves fidelity, DanceGRPO exhibits noticeable visual artifacts.

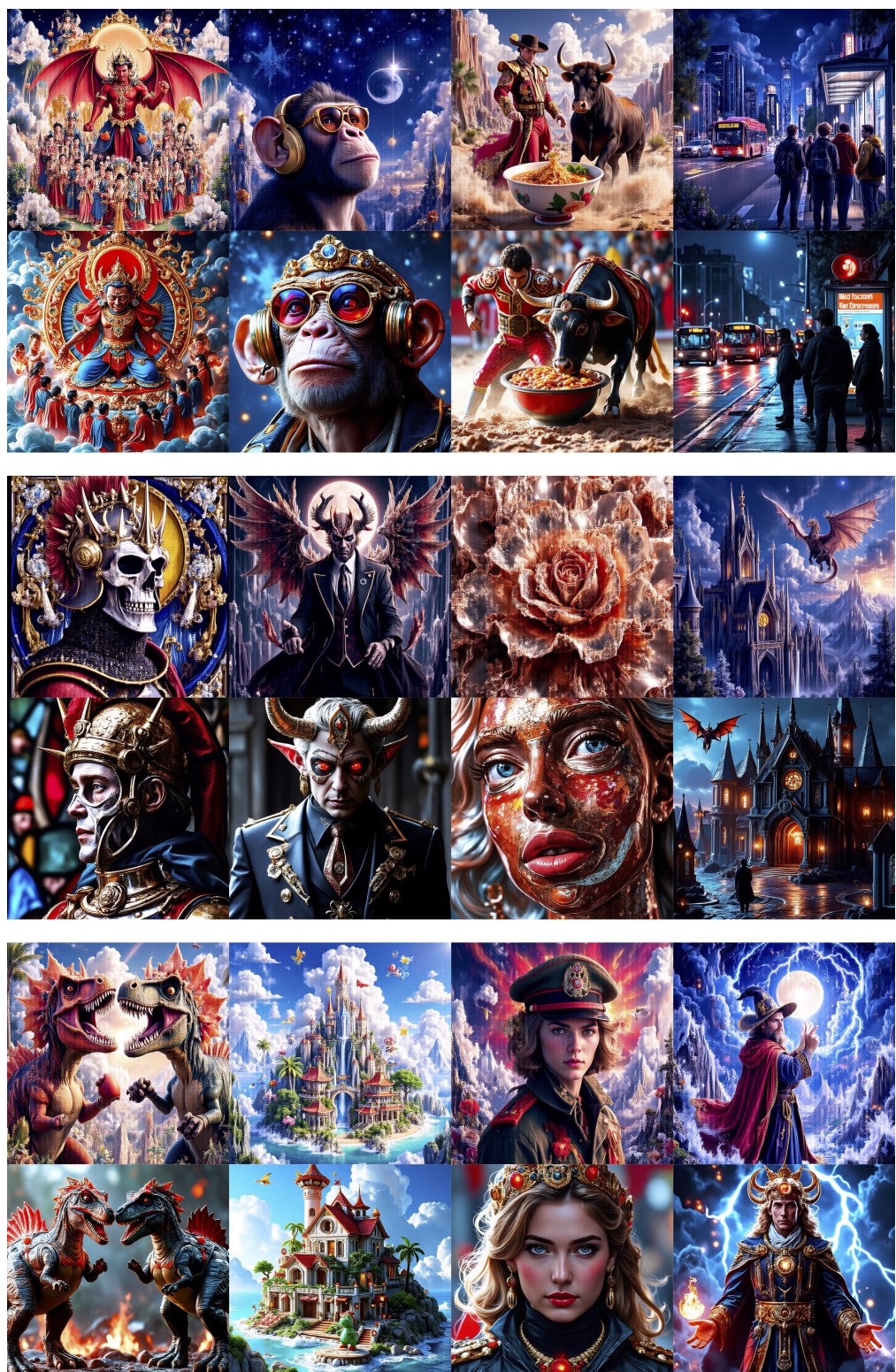

Figure 20: **PCPO is more robust to quality degradation from prolonged training.** A comparison at epoch 240 on the FLUX model. While extensive training for minimal reward gains causes the baseline (top row) to suffer significant image degradation and artifacts, PCPO (bottom row) maintains its quality much more effectively.

# I    USE OF LARGE LANGUAGE MODELS

We utilized Large Language Models (LLMs) as assistive tools throughout the research and manuscript preparation process. For research ideation, Google's Gemini was instrumental in suggesting the use of LMMs to control for prompt-level variance in our analysis. OpenAI's ChatGPT, Github's Copilot, and Gemini were employed to help draft and refine code implementations. In preparing the manuscript, ChatGPT and Gemini were employed to compose initial paragraph drafts from detailed research notes. In addition, these LLMs assisted in restructuring sentences for improved readability.

