# OpenReview forum: "PCPO: Proportionate Credit Policy Optimization for Preference Alignment of Image Generation Models"
_ICLR.cc/2026/Conference — ICLR 2026 Poster_

### Official Review · Reviewer_gCoT · 2025-10-16

**Soundness:** 2
**Presentation:** 3
**Contribution:** 2
**Rating:** 4
**Confidence:** 3

**Summary:**

This paper proposes Proportionate Credit Policy Optimization (PCPO), a reinforcement learning framework for aligning text-to-image (T2I) generation models. The authors identify disproportionate credit assignment across diffusion or flow timesteps as a core source of instability in PPO/GRPO-style methods. This issue leads to high-variance gradients, excessive clipping, and eventual model collapse. Empirical results across diffusion and flow models: Stable Diffusion 1.4/1.5 and FLUX, demonstrate: faster convergence, improved image fidelity, and reduced model collapse.

**Strengths:**

1. **Quality**

Ablations convincingly demonstrate the cumulative contribution of PCPO’s components. Detailed appendices cover proofs, hyperparameters, and resource requirements for reproducibility.


2. **Experimental Rigor**

The inclusion of Linear Mixed Model (LMM) statistical analyses adds robustness to the evaluation, while the human preference study further validates perceptual improvements.


3. **Clarity**

The paper is well-organized and clearly written, despite the technical density. Visualizations, e.g., Figures 2–4, effectively illustrate instability sources and PCPO’s corrective mechanism. Mathematical derivations are complete and logically consistent.

**Weaknesses:**

1. **Theoretical Intuition.**

While the proportionality argument is sound, the paper could better connect it to established temporal credit assignment or variance reduction principles in RL.


2. **Lack of baselines.**

It is encouraged to compare with other SOTA methods. See Q3.

While results on SD1.4/1.5 and FLUX are compelling, broader testing, e.g., on SDXL (Podell et al., 2023), is recommended to establish generality.

**Questions:**

1. **Computation vs. Speedup.** It reports per-epoch acceleration, but omits detailed wall-clock comparisons, leaving open questions about added per-step computation.

2. **Variance Quantification.**
Beyond the clipping fraction, could the paper include explicit variance or gradient norm distributions to quantify the stability gains?

3. **Baselines.** More baselines are encouraged to compare with, such as SPPO [1] and RainbowPA [2]. On the other hand, can PCPO be integrated with reward-model-free methods (e.g., self-play [3]) to further stabilize alignment training?

4. $\log \rho_{t} \approx \rho_{t} - 1$ only holds when $\rho_{t}$ is about $1$. This may fail in high-variance regions or off-policy updates, especially when sampling trajectories from older policies or noisy prompts. Could you please show the distribution of $\rho_{t}$?


[1] Bridging SFT and DPO for Diffusion Model Alignment with Self-Sampling Preference Optimization. arXiv:2410.05255, 2025.

[2] Diffusion-RainbowPA: Improvements Integrated Preference Alignment for Diffusion-based Text-to-Image Generation. Transactions on Machine Learning Research, 2025.

[3] Self-Play Fine-Tuning of Diffusion Models for Text-to-Image Generation. NeurIPS 2024.

---

> ### Author Response · Authors · 2025-11-20
>
> We sincerely thank the reviewer for their thorough and insightful review.  We have worked to address every constructive criticism in our revision, which we believe has substantially strengthened the paper.
>
> ### Response to Weaknesses
>
> **W1: Theoretical Intuition**
>
> > "While the proportionality argument is sound, the paper could better connect it to established temporal credit assignment or variance reduction principles in RL."
>
> We agree that a stronger connection to foundational RL principles strengthens our justification. In our revision, we have added Appendix E, which draws a direct analogy between our proportionality principle and the foundational REINFORCE algorithm [1].
>
> We show that the native sampler weights $w(t)$ act as arbitrary, non-proportional scalers on what is analogous to REINFORCE's "eligibility vector". This arbitrary scaling, an artifact of the sampler's math, is what perturbs credit assignment. PCPO corrects this by reweighting to ensure credit is assigned proportionally to the timestep interval $\Delta t$, restoring the sound credit assignment principle inherent in REINFORCE.
>
> **W2: Lack of Baselines & Generality**
>
> > "It is encouraged to compare with other SOTA methods... broader testing, e.g., on SDXL... is recommended to establish generality."
>
> We have expanded our experiments to address both SOTA comparisons and generality.
>
> * **Generality on SOTA Models:** We added new experiments applying PCPO to the Flow-GRPO framework on the SD3.5-M model (Figure 5). As shown in our new results, PCPO robustly outperforms the Flow-GRPO baseline in this entirely different training regime (new backbone model, new noise schedule, new rewards, $\beta_{KL}$ penalty), confirming its generalizability beyond SD1.x and FLUX.
> * **SOTA Method Baselines:** As requested in Q3, we have also added a direct comparison against SOTA DPO and self-play methods in Appendix C. We detail this in our response to Q3 below.

---

> ### Author Response · Authors · 2025-11-20
>
> ### Response to Questions
>
> **1. Computation vs. Speedup**
>
> > "It reports per-epoch acceleration, but omits detailed wall-clock comparisons, leaving open questions about added per-step computation."
>
> This is a critical point. We have added **Wall-Clock Time and Computational Cost** in  **Appendix D** to clarify this.
>
> * **No Overhead:** As shown in our new wall-clock analysis (Figure 9), PCPO adds no computational overhead and is, in fact, slightly faster per epoch than the baseline. This is because PCPO does not require the additional reverse SDE sampler step to compute $p_\theta$ for policy updates.
>
> Therefore, the reported per-epoch speedups directly translate to reduced wall-clock time.
>
> **2. Variance Quantification**
>
> > "Beyond the clipping fraction, could the paper include explicit variance or gradient norm distributions to quantify the stability gains?"
>
> This is an excellent suggestion. To provide a more direct measure of stability, we have added a quantitative gradient analysis in **Gradient Stability Analysis** in **Appendix D**.
>
> Figure 11 shows $\mathbb{E}[|g_t|]$ and $\text{Var}[|g_t|]$ of the first-layer, measured at both the start of training and near convergence. It shows that PCPO's gradients remain more stable, especially near convergence.
>
> **3. Baselines (SPPO, RainbowPA, Self-Play)**
>
> > "More baselines are encouraged to compare with... can PCPO be integrated with reward-model-free methods..."
>
> We have added a new section, **Appendix C**, comparing PCPO to SOTA DPO (LPO [2]) and self-play (SSPO [3], SPIN-Diffusion [4]) methods. To the best of our knowledge, the methods suggested by the reviewer lack open weights. As such, we cite results from the corresponding papers. RainbowPA [5] did not report validation results on Pick-a-Pic prompts, making a head-to-head comparison difficult.
>
> * **Setup:** It is difficult to fairly compare reward-free methods against reward-based methods. Following prior work [2, 6], we set an experimental protocol in favor of reward-free methods, by training on Pick-a-Pic prompts while restricting reward model choice to the PickScore reward. As absolute scores vary across papers (due to different validation subsets), we compare the performance gains over the base model, as reported in their work.
> * **Results:** Table 10 shows that PCPO achieves superior gains across Aesthetics Score, PickScore, and ImageReward. While it lags behind SSPO in HPSv2 gains, we note that SSPO was trained on Pick-a-Pic v2 prompts, a richer dataset. Moreover, SSPO reports an atypically low performance for the base model's performance on HPSv2 (0.238 compared to 0.261, 0.271, 0.265 of other works), which suggests that their validation set was particularly challenging for the base model and hence inflating relative gain.
>
> PCPO was designed as an improvement to policy gradient (PPO, GRPO) based methods, so it is not directly applicable to reward-free methods. However, the insights of our work could transfer to different regimes, by analyzing discrepancies between contribution and credition.
>
> We add that work on stabilizing gradients for DPO methods have focused on the disproportionate ratio between winning and losing gradients, mainly in the LLM domain [7]. Future work could provide a unifying perspective on gradient stability for alignment training in general.

---

> ### Author Response · Authors · 2025-11-20
>
> **4. Soundness of $\log \rho_t \approx \rho_t - 1$ Approximation**
>
> > "This may fail in high-variance regions or off-policy updates... Could you please show the distribution of $\rho$?"
>
> We have added a detailed justification in **Soundness of $\log \rho_t \approx \rho_t - 1$ Approximation** in **Appendix D** that addresses this from four perspectives:
>
> 1.  **Empirical Distribution:** We explicitly plotted the distribution of $\rho_t$ during training, and verified that its value remains exceptionally close to 1.0 (mean $\approx 1.0$, std $\approx 10^{-4}$).
> 2. **Error Bounding:** The PPO clipping mechanism itself bounds the effective approximation error. The gradient only flows when $|\log \rho_t| < \xi$ (where $\xi = 10^{-4}$). In this tiny region, the Taylor error is negligible (on the order of $\mathcal{O}(\xi^2) \sim 10^{-8}$).
> 3.  **Numerical Stability:** We argue that the $\rho_t = \exp(\log \rho_t)$ computation in standard PPO is a larger source of error (floating-point instability, $\sim 10^{-5}$), which PCPO avoids by operating directly in log-space. Note that precision errors, including the floating-point instability of the $\exp(\cdot)$ operation, is a practical issue that forces the usage of same batch sizes for sampling and policy updates [8]. This makes the sampling phase inefficient, as it can typically fit a $2 \times$ larger batch size into VRAM since it does not require storing gradients.
> 4.  **Theoretical Soundness:** Finally, as shown by [9], the $\rho_t - 1$ term acts as a swappable "classifier" in a hinge loss. Our $\log \rho_t$ objective is sound on its own merits, independent of the approximation.
>
> ---
>
> We thank the reviewer again for the valuable feedback, and hope these revisions address the reviewer's concerns.
>
> ---
> ### References
> [1] Richard S. Sutton and Andrew G. Barto. Reinforcement Learning: An Introduction. The MIT Press, second edition, 2018. URL http://incompleteideas.net/book/the-book-2nd.html
>
> [2] Tao Zhang, Cheng Da, Kun Ding, Kun Jin, Yan Li, Tingting Gao, Di Zhang, Shiming Xiang, and Chunhong Pan. Diffusion model as a noise-aware latent reward model for step-level preference optimization. arXiv preprint arXiv:2502.01051, 2025b.
>
> [3] Daoan Zhang, Guangchen Lan, Dong-Jun Han, Wenlin Yao, Xiaoman Pan, Hongming Zhang, Mingxiao Li, Pengcheng Chen, Yu Dong, Christopher Brinton, and Jiebo Luo. Bridging sft and dpo for diffusion model alignment with self-sampling preference optimization, 2025a. URL https://arxiv.org/abs/2410.05255
>
> [4] Huizhuo Yuan, Zixiang Chen, Kaixuan Ji, and Quanquan Gu. Self-play fine-tuning of diffusion models for text-to-image generation. In Proceedings of the 38th International Conference on Neural Information Processing Systems, NIPS ’24, Red Hook, NY, USA, 2024. Curran Associates Inc. ISBN 9798331314385.
>
> [5] Haoyuan Sun, Bin Liang, Bo Xia, Jiaqi Wu, Yifei Zhao, Kai Qin, Yongzhe Chang, and Xueqian Wang. Diffusion-RainbowPA: Improvements Integrated Preference Alignment for Diffusion-based Text-to-Image Generation. Transactions on Machine Learning Research, 2025.
>
> [6] Zhanhao Liang, Yuhui Yuan, Shuyang Gu, Bohan Chen, Tiankai Hang, Mingxi Cheng, Ji Li, and Liang Zheng. Aesthetic post-training diffusion models from generic preferences with step-by-step preference optimization. In 2025 IEEE/CVF Conference on Computer Vision and Pattern Recognition (CVPR), pp. 13199–13208, 2025. doi: 10.1109/CVPR52734.2025.01232.
>
> [7] Qinwei Ma, Jingzhe Shi, Can Jin, Jenq-Neng Hwang, Serge Belongie, Lei Li. Gradient Imbalance in Direct Preference Optimization, 2025. URL https://arxiv.org/abs/2502.20847
>
> [8] Yash Savani, Jie Liu, and Jiexuanz. Gradient accumulation and batch size handling in training loop, 2025. URL https://github.com/yifan123/flow_grpo/issues/73#issuecomment-3172334321. Accessed: 2025-11-20.
>
> [9] Nai-Chieh Huang, Ping-Chun Hsieh, Kuo-Hao Ho, and I-Chen Wu. Ppo-clip attains global optimality: Towards deeper understandings of clipping. In AAAI, pp. 12600–12607, 2024.

---

### Official Review · Reviewer_oEo3 · 2025-11-01

**Soundness:** 3
**Presentation:** 3
**Contribution:** 3
**Rating:** 6
**Confidence:** 3

**Summary:**

This paper proposes PCPO (Proportionate Credit Preference Optimization), a new framework to address the credit assignment problem in diffusion model preference optimization. While previous methods (e.g., DPO, LPO, TAPO) apply uniform or stepwise weighting across diffusion steps, they fail to proportionally assign reward signals according to the relative contribution of each latent step to the final image quality.
PCPO introduces a proportionate credit weighting mechanism that computes latent-wise contributions via gradient-based sensitivity analysis, enabling consistent and efficient optimization across diffusion trajectories. The authors provide a theoretical justification linking PCPO to policy gradient credit assignment and demonstrate its empirical advantage on text-to-image (SDXL, SD3.5) and text-to-video (ModelScope) benchmarks, improving both alignment and fidelity metrics.

**Strengths:**

1. Novel problem formulation: The paper clearly identifies the overlooked issue of disproportionate credit assignment in trajectory-based diffusion alignment.
2. Technically novelty: The gradient-based proportional credit estimation is simple yet effective, bridging reinforcement learning and diffusion optimization perspectives.
3. Strong empirical validation: Experiments on multiple backbones (SDXL, SD3.5, ModelScope) show consistent improvements over DPO, LPO, and TAPO.

**Weaknesses:**

1. Missing comparisons on metrics such as HPS, CLIP, Pick-a-Pic(Not reward), etc. FID and IS is not enough to evaluate the model quality.
2.  Sensitivity & hyperparameters: Sensitivity plots exist but more systematic sweeps (schedules, batch sizes, guidance scales, LoRA ranks, SDE noise levels) can strengthen claims.

**Questions:**

1. Add more metrics?
2. Add more ablations?

---

> ### Author Response · Authors · 2025-11-20
>
> We thank the reviewer for the positive assessment of our work, and  appreciate the constructive feedback. We report results on additional results to address the reviewer's concerns.
>
> ### Response to Weaknesses
>
> **W1: Missing comparisons on metrics (HPS, CLIP, Pick-a-Pic, etc.)**
>
> > "Missing comparisons on metrics such as HPS, CLIP, Pick-a-Pic(Not reward), etc. FID and IS is not enough to evaluate the model quality."
>
> We agree that FID and IS alone are insufficient for a comprehensive evaluation of alignment. In our revision, we have added **Evaluation with Unseen Prompts on Diverse Reward Metrics** in **Section 3.2**.
>
> 1.  **New Metrics:** We now evaluate our models on a diverse suite of alignment metrics: HPSv2.1, Aesthetics, CLIPScore, PickScore, and ImageReward.
>
> 2.  **Unseen Prompts:** To test robustness, these metrics are calculated on held-out prompt sets: **MSCOCO-2017** (5K validation) and **MJHQ-30K** (5K subset).
>
> 3.  **Results:** As shown in **Table 5**, PCPO outperforms the DanceGRPO baseline. This trend is agnostic of generation scheme (diffusion/flow) or metric. Notably, PCPO maintains better text-image alignment (**CLIPScore**) than the baseline, especiallly for the relatively complex MJHQ dataset, suggesting it is less prone to reward hacking.
>
> 4.  **FD_DINOv2:** We also added Fréchet Distance using DINOv2 features ($\text{FD}_\text{DINO}$) in **Table 2** to provide a more modern semantic measure of distributional drift.
>
>
> **W2: Sensitivity & Hyperparameters**
>
> > "Sensitivity & hyperparameters: Sensitivity plots exist but more systematic sweeps... can strengthen claims."
>
> To demonstrate that PCPO is robust and not dependent on a narrow set of hyperparameters, we have expanded our evaluation to cover **10 distinct training configurations** (summarized in **Table 6** and **Table 7**).
>
> 1.  **Generalization to SOTA Architectures (SD3.5-M):** We applied PCPO to the **SD3.5-M** model using the **Flow-GRPO** framework (Configs 8 & 9). This exhibits a drastic change in settings and hyperparameters, including a **new model** (SD3.5-M), **new noise schedule** (Flow-GRPO SDE), **new rewards** (OCR, PickScore), **new objective** (auxiliary KL penalty), and **new prompts** (OCR, Pick-a-Pic prompts).
>
> 2.  **Results:** Despite these changes, PCPO consistently outperformed the Flow-GRPO baseline. **Figure 5** shows that  PCPO achieves higher rewards with lower KL divergence and significantly reduced clipping.
>
> Our results now span **3 baseline algorithms** (DDPO, DanceGRPO, Flow-GRPO), **4 base models** (SD1.4, SD1.5, FLUX.1-dev, SD3.5-M), and **5 reward functions**. PCPO provides consistent gains across all these settings, confirming that our proportionality principle is a fundamental correction rather than a hyperparameter-sensitive heuristic.

---

> ### Author Response · Authors · 2025-11-20
>
> ### Response to Questions
>
> **Q1: Add more metrics?**
>
> Please view our response to Weakness 1.
>
> **Q2: Add more ablations?**
>
> To provide deeper mechanical insight into _how_ PCPO stabilizes training, we added a quantitative **Gradient Stability Analysis** in **Appendix D**.
>
> -   **Gradient Variance:** In **Figure 11**, we plot $\mathbb{E}[∣g_t∣]$ and $\text{Var}(∣g_t∣)$ in the first layer. The results show that while PCPO maintains significantly more stable gradients, especially near convergence.
>
> To strengthen our justificaiton, we also added **Soundness of $\log \rho_t \approx \rho_t - 1$ Approximation** in **Appendix D**.
>
> -   **Ratio Distribution:** We plotted the empirical distribution of $\rho_t$ (**Figure 10**), confirming that our updates stay within the region where the Taylor approximation is highly accurate. We also elaborate three additional aspects that support the validity of the Taylor approximation.
>
> ---
> We hope these additional experiments and analyses address the reviewer's concerns.

---

### Official Review · Reviewer_71VF · 2025-11-06

**Soundness:** 3
**Presentation:** 2
**Contribution:** 2
**Rating:** 4
**Confidence:** 3

**Summary:**

This paper reveals that disproportionate credit assignment is  a key reason of instability when training T2I models using reinforcement learning. Proportionate Credit Policy Optimization (PCPO) is therefore introduced to deal with this problem. PCPO improve training stability of RL in training T2I models, leading to convergence acceleration and superior image quality.

**Strengths:**

1. This paper proposes Proportionate Credit Policy Optimization to deal with  training instability of RL for T2I fine-tuning. PCPO not only enhances numerical stability, but also ensures proportional credit assignment. The author also provide detailed theoretical derivation.

2. Compared to DanceGRPO, PCPO introduces  convergence acceleration and superior image quality.

**Weaknesses:**

1. It would be beneficial to include an introductory overview of diffusion models and GRPO in the preliminary section, as their absence hinders the paper's accessibility for researchers new to the field.

2. FID can not measure the human preference score of the generated images and text-image consistency, CLIP Score, ImageReward, and HPSv2  should be included in the  Tables  for comparison

3. The text condition for image generation in Figure 13 and 14 is too simple,  I would like to see more complex qualitative comparisons in Appendix G, and text condition should be given so that I can observe text-image alignment.

4. The experiment is not conducted on widely used datasets, e.g., MSCOCO-2014 30K validation  dataset, MSCOCO-2017 5K validation  dataset, MJHQ-30K dataset. It is benefical to convince me via reporting the results on above datasets.

**Questions:**

Table 1 employs only a single target level for each reward. It is important to investigate whether the reported conclusions remain consistent across different target levels. A discussion on the robustness and generalizability of the results with respect to this parameter would strengthen the paper.

---

> ### Author Response · Authors · 2025-11-20
>
> We thank the reviewer for constructive feedback. We have revised the paper to address your concerns.
>
> ### Response to Weaknesses
>
> **W1: Missing overview of diffusion models and GRPO**
>
> > "It would be beneficial to include an introductory overview of diffusion models and GRPO in the preliminary section..."
>
> We agree that this improves accessibility. We have expanded **Section 2.1 (Preliminaries)** to include a dedicated overview of **Diffusion and Flow Matching** mechanics, and briefly introduced the GRPO framework in **Policy Gradient Alignment**.
>
> **W2, W4: Need more metrics (CLIP, ImageReward, HPSv2), Need experiments on widely used datasets (MSCOCO, MJHQ)**
>
> > "FID can not measure the human preference score... CLIP Score, ImageReward, and HPSv2 should be included..."
> >
> > "The experiment is not conducted on widely used datasets, e.g., MSCOCO... MJHQ-30K..."
>
> In **Table 5** in **Section 3.2**, we have added a comprehensive evaluation using HPSv2.1, CLIPScore, PickScore, ImageReward, and Aesthetics on the MSCOCO-2017 5K validation and MJHQ-30K datasets. PCPO achieves superior performance across all metrics compared to the DanceGRPO baseline. Notably, PCPO is better at maintaining CLIPScore while optimizing for preference rewards, indicating better resistance to reward hacking.
>
> **W3: Simple text conditions in qualitative comparisons**
>
> > "The text condition... in Figure 13 and 14 is too simple... I would like to see more complex qualitative comparisons..."
>
> We clarify that Figures 13 and 14 (Figures 15 and 17 in the revised paper) utilize the **DDPO** framework, which is known to struggle with complex, open-vocabulary prompts due to its  specific fine-tuning setup (as detailed in *Limitations* of **Appendix B.2.1**). Analogous to how PPO-based LLM fine-tuning failed to unlock reasoning capabilities, PPO-based fine-tuning of diffusion models (DDPO) lacks more advanced capabilites accessible to GRPO-based fine-tuning (DanceGRPO, SD1.x).
>
> To address your request for qualitative comparisons on generalization to other prompts, we direct your attention to Figure 16, where we provide samples generated on unseen prompts, akin to Figure 15 in the original DDPO work [1]. Note that, however, both checkpoints have very limited generalization capabilites compared to the base model.

---

> ### Author Response · Authors · 2025-11-20
>
> ### Response to Questions
>
> **Q1: Robustness across different target levels**
>
> > "Table 1 employs only a single target level... investigate whether the reported conclusions remain consistent across different target levels."
>
> We kindly remind the reviewer that we have already provided the full breakdown of speedups across **multiple target reward levels** in **Table 10 (Appendix D)**, or Table 6 in the previous version. Results quantify PCPO's consistent acceleration throughout training across various thresholds. We have updated the main text references to explicitly point readers to Table 10 for this full breakdown.
>
> ---
>
> We believe these revisions directly address the weaknesses identified by the reviewer.
>
> ### References
> [1] Kevin Black, Michael Janner, Yilun Du, Ilya Kostrikov, and Sergey Levine. Training diffusion models with reinforcement learning. In The Twelfth International Conference on Learning Representations, 2024. URL https://openreview.net/forum?id=YCWjhGrJFD.

---

### Official Review · Reviewer_4hys · 2025-11-08

**Soundness:** 3
**Presentation:** 3
**Contribution:** 2
**Rating:** 8
**Confidence:** 3

**Summary:**

The paper identifies disproportionate credit assignment across timesteps as a major cause of instability in the alignment for text-to-image models.
It proposes Proportionate Credit Policy Optimization (PCPO), which reformulates the PPO objective into a numerically stable log-hinge view and then enforces proportional credit over the sampling trajectory.
For diffusion models, PCPO makes per-step weights uniform by improving the variance schedule; for flow models, it restores proportionality by reweighting the training loss by each step’s integration interval, avoiding invasive sampler changes.

Across DDPO and DanceGRPO, PCPO reduces clipping and variance, speeds convergence, and improves fidelity, as indicated by lower FIDs, while mitigating collapse.
A small human-preference study favors PCPO over strong baselines. Overall, the method offers a simple fix that stabilizes training and yields samples of better fidelity at similar reward levels.

**Strengths:**

- Clarity. This paper is well written in terms of motivation and technical content. The paper clearly states the core problem, training instability in T2I alignment due to disproportionate credit across timesteps. The motivation and scope easy to follow. Figures helps understand the volatile and modified weighting clearly.
- Technical contributions. Given the replace of $\rho_t-1$ with $\log \rho_t$, the motivation is clear. This work further provides fix to the weighting under time shift for diffusion and flow models.
- Improvements. PCPO offers convergence acceleration to the target reward levels from 24.6% to 41.2%, while keeping better fidelity, as indicated by FID scores.

**Weaknesses:**

- Metrics. FID provides a reasonable measurement of the shift from the base distribution, though it is less accurate for images of higher resolutions. It would be better to have modern metrics such as FD-DINOv2 for fidelity evaluation.
- Regarding IS, it would be more scientific to claim that given similar FID levels, a higher IS indicates better quality; or given a similar IS score, a lower FID indicates less mode collapse.

To be clear, this paper is of good quality regarding a detailed question in the alignment of diffusion and flow models, and meets the bar of ICLR. The following questions and suggestions could improve the quality.

Suggestions:
- Plot per‑timestep gradient contributions and clipping rates before/after PCPO; report variance reduction in those contributions.
- Quantify the Taylor error during training with a time‑series plot; show whether the stability gain comes primarily from the proportionality fix or from the log‑hinge.

Typos:
- model collapse -> mode collapse

**Questions:**

- Could you please report the $E[\|g_t\|]$, $Var(\|g_t\|)$, and clip rate w.r.t. t for basline and PCPO? Having two visualizations/reports in the early stage, and near convergence, would be better.
- Why does it cause a negative effect when over-weighting the higher noise region?

---

> ### Author Response · Authors · 2025-11-20
>
> We sincerely thank the reviewer for the encouraging assessment of our work. We have incorporated your suggestions into the revision.
>
> ### Response to Weaknesses
>
> **W1: Modern Metrics (FD-DINOv2)**
>
> > "It would be better to have modern metrics such as FD-DINOv2 for fidelity evaluation."
>
> We fully agree. We have added **Fréchet Distance DINOv2 ($\textbf{FD}_\textbf{DINO}$)** to our evaluation in **Table 2** (and **Table 3, 4** for LMM analysis).
>
> -   **Results:** PCPO achieves lower (better) FD-DINO scores across all experiments. Most notably, for the FLUX model, PCPO achieves a statistically significant improvement over the baseline (p=0.001).
>
> -   **Cross-Reward Validation:** To further ensure robustness beyond standard metrics, we added **Table 5**, evaluating on various metrics on unseen prompts. PCPO consistently outperforms the baseline here as well.
>
>
> **W2: Interpretation of Inception Score (IS)**
>
> > "It would be more scientific to claim that... given a similar IS score, a lower FID indicates less mode collapse."
>
> We have revised **Section 3.2** to clarify our interpretation of IS in the context of RL fine-tuning.
>
> -   **Context:** As observed in the ICLR spotlight paper **CADS** [1], models can artificially inflate IS by collapsing into a few high-confidence modes.
>
> -   **Our Finding:** We empirically found that increasing batch size (a known strategy to reduce statistical approximation error, leading to mitigation of model collapse [2]) _decreases_ IS ($\beta_\text{batch}=−0.266, p=0.011$).
>
> -   **Revised Claim:** Therefore, we argue that **in this specific fine-tuning context**, a lower IS, **when accompanied by a lower or comparable FID**, indicates successful mitigation of mode collapse and better preservation of the base distribution.
>
> If this revised claim still raises concerns, please let us know.

---

> ### Author Response · Authors · 2025-11-20
>
> ### Response to Suggestions
>
> **S1: Plot per-timestep gradient contributions**
>
> > "Plot per‑timestep gradient contributions and clipping rates..."
>
> We have added this analysis in **Gradient Stability Analysis** in **Appendix D**. Figure 11 plots the $\mathbb{E}[∣g_t∣]$ and  $\text{Var}(∣g_t∣)$ of the gradients.
>
> -   **Result:** The plots show that PCPO maintains significantly more stable gradients throughout training, especially near convergence.
>
>
> **S2: Quantify Taylor Error**
>
> > "Quantify the Taylor error... show whether the stability gain comes primarily from the proportionality fix or from the log‑hinge."
>
> We addressed this in **Soundness of $\log \rho_t \approx \rho_t - 1$ Approximation** in **Appendix D**.
>
> -   **Quantification:** In Figure 10, we plotted the empirical distribution of $\rho_t$. The deviation $∣\rho_t−1∣$ is consistently on the order of $10^{−3}$ or less. Since the Taylor error is second-order, the error is negligible ($\sim 10^{-6}$). We also provide further justification based on the clipping mechanism, numerical stability of avoiding exponentials, and the work of [7].
>
> -   **Ablation:** Our ablation study (**Figure 6**) explicitly separates these factors. It shows that while the log-hinge offers some stability (green line), the **proportionality fix** (orange line) is the primary, decisive factor for reducing clipping and accelerating training.
>
> ### Response to Questions
>
> **Q1: Report stats w.r.t t in early stage and near convergence?**
>
> > "Could you please report the... clip rate w.r.t. t for basline and PCPO?"
>
> Please see our reply to Suggestion 1.
>
> **Q2: Why does over-weighting high noise cause negative effects?**
>
> > "Why does it cause a negative effect when over-weighting the higher noise region?"
>
> Naive overweighting (as tested in **Figure 7d**) distorts the reward signal. If a lower-noise step contributed significantly to the final image quality, but we arbitrarily overweight the high-noise steps, we effectively "blame" or "credit" the wrong timesteps.
>
> The effects of spurious reward signals have been studied in the LLM domain. Random or spurious rewards can yield reward gains, but it comes at the cost of entropy reduction [3, 4]. Even in the LLM domain, the underlying mechanisms for this phenomenon is understood only through a mix of theory and empirical observations [4]. For image generative models, our intuition is that it allows for more *value function bias*, i.e. biasing the fine-tuned model away from the desired *tilted distribution* $p^\text{base}(x) \exp(r(x))$ and introducing more dependency on initial noise [5].
>
> While focusing on high-noise regions can be beneficial (as they often contain more semantic changes), our results suggest that it must be done through principled methods such as performing **trajectory branching** (i.e., suggested by **TempFlow-GRPO** [6]) more frequently at high-noise timesteps.
>
> ### Typos
>
> > "model collapse -> mode collapse"
>
> We use "model collapse" as an umbrella term to cover both diversity loss and quality degradation [2]. However, we have updated the text to use "mode collapse" specifically when referring to diversity loss to ensure precision. We have expanded the Related Work section to explain our usage of this term.
>
> ---
> We thank you again for your detailed review, which helped us significantly strengthen our work.

---

> ### Author Response · Authors · 2025-11-20
>
> ### References
> [1] Seyedmorteza Sadat, Jakob Buhmann, Derek Bradley, Otmar Hilliges, and Romann M. Weber. CADS: Unleashing the diversity of diffusion models through condition-annealed sampling. In The Twelfth International Conference on Learning Representations, 2024. URL https://openreview.net/forum?id=zMoNrajk2X.
>
> [2] Ilia Shumailov, Zakhar Shumaylov, Yiren Zhao, Nicolas Papernot, Ross Anderson, and Yarin Gal. Ai models collapse when trained on recursively generated data. Nature, 631(8022):755–759, jul 2024. ISSN 1476-4687. doi: 10.1038/s41586-024-07566-y. URL https://doi.org/10.1038/s41586-024-07566-y.
>
> [3] Rulin Shao, Shuyue Stella Li, Rui Xin, Scott Geng, Yiping Wang, Sewoong Oh, Simon Shaolei Du, Nathan Lambert, Sewon Min, Ranjay Krishna, Yulia Tsvetkov, Hannaneh Hajishirzi, Pang Wei Koh, and Luke Zettlemoyer. Spurious Rewards: Rethinking Training Signals in RLVR, 2025. URL https://arxiv.org/abs/2506.10947.
>
> [4] Jaesung R. Park, Junsu Kim, Gyeongman Kim, Jinyoung Jo, Sean Choi, Jaewoong Cho, and Ernest K. Ryu. Clip-low increases entropy and clip-high decreases entropy in reinforcement learning of large language models, 2025. URL https://arxiv.org/abs/2509.26114.
>
> [5] Carles Domingo-Enrich, Michal Drozdzal, Brian Karrer, and Ricky T. Q. Chen. Adjoint Matching: Fine-tuning Flow and Diffusion Generative Models with Memoryless Stochastic Optimal Control, 2024. URL https://arxiv.org/abs/2409.08861.
>
> [6] Xiaoxuan He, Siming Fu, Yuke Zhao, Wanli Li, Jian Yang, Dacheng Yin, Fengyun Rao, and Bo Zhang. Tempflow-grpo: When timing matters for grpo in flow models, 2025. URL https://arxiv.org/abs/2508.04324.
>
> [7] Nai-Chieh Huang, Ping-Chun Hsieh, Kuo-Hao Ho, and I-Chen Wu. Ppo-clip attains global optimality: Towards deeper understandings of clipping. In AAAI, pp. 12600–12607, 2024.

---

### Author Response · Authors · 2025-11-20

### Supplementary Material: Anonymized Core Implementation

Dear Reviewers and Area Chair,

In our Reproducibility Statement, we committed to open-sourcing the code during the review period. To ensure strict compliance with double-blind reviewing policies and prevent any potential de-anonymization risks (e.g., absolute paths and server-specific logging), we have opted to release the core implementation files rather than the full code.

This repository isolates the key algorithmic components of PCPO, allowing for a direct and clear comparison of the logic differences between our method and the baselines.

Anonymized Repository: https://anonymous.4open.science/r/pcpo-0F1C

Please note that the auxiliary helper functions and data loaders not included in this repository are shared with the standard implementations. For those components, please refer to the official codebases of the respective baselines (DDPO [1], DanceGRPO [2], and Flow-GRPO [3]).

We believe this focused release allows for the most efficient verification of our technical contributions while maintaining the integrity of the double-blind process.

### References
[1] Kevin Black. ddpo-pytorch. URL https://github.com/kvablack/ddpo-pytorch. Accessed: 2025-11-20.

[2] Zeyue Xue. DanceGRPO. URL https://github.com/XueZeyue/DanceGRPO. Accessed: 2025-11-20.

[3] Jie Liu. flow_grpo. URL https://github.com/yifan123/flow_grpo. Accessed: 2025-11-20.

---

### Author Response · Authors · 2025-11-20
**General Response**

We sincerely thank the reviewers for their constructive and detailed feedback. We are encouraged by the consensus on our **novelty and technical contribution** (`4hys`, `oEo3`),  **effectiveness of our solution** (`4hys`, `71VF`,  `oEo3`, `gCoT`), and **clarity** (`4hys`, `gCoT`).

In response to the valuable suggestions regarding generalization, evaluation rigor, and theoretical intuition, we have performed extensive new experiments and analyses. Below is a summary of the major revisions included in the updated manuscript.

**1. Generalization.** To address concerns regarding generality beyond SD1.x/FLUX, we expanded our evaluation to include **10 distinct training configurations** (Table 7).

-   **New Model & Framework:** We applied PCPO to **Stable Diffusion 3.5-Medium** using the **Flow-GRPO** framework. This setup involves a completely different noise schedule, new rewards (OCR, PickScore), and an auxiliary KL penalty.

-   **Result:** Despite these drastic changes, PCPO robustly outperforms the baseline (Figure 5), confirming that our proportionality principle is a fundamental correction applicable to diverse generative architectures.


**2. Comprehensive Evaluation with Modern Metrics.** We agree that FID/IS alone were insufficient. We have added a rigorous evaluation suite across **5 reward metrics** on **unseen prompts** (MSCOCO-2017 and MJHQ-30K):

-   **Metrics:** HPSv2.1, Aesthetics, CLIPScore, PickScore, and ImageReward.

-   **Fidelity:** We also added **Fréchet Distance DINOv2 (FD-DINO​)** to provide a modern semantic measure of distributional drift.

-   **Result:** PCPO consistently achieves superior alignment (Table 5) and fidelity scores (Table 2) compared to DanceGRPO across unseen datasets, demonstrating better generalization and resistance to reward hacking.


**3. Deepened Theoretical & Mechanistic Analysis** To provide concrete evidence of _why_ PCPO stabilizes training, we added two key analyses in **Appendix D**:

-   **Approximation Soundness:** We empirically verified that the policy ratio $\rho_t$ remains exceptionally close to 1.0 (Figure 10), ensuring the Taylor approximation error ($\log \rho_t​ \approx \rho_t​ − 1$) is negligible. We also demonstrated that the clipping mechanism strictly bounds this error whenever gradients are active.

-   **Gradient Stability:** We visualized the gradient distribution of the first layer ($\mathbb{E}[|g_t|]$ and $\text{Var}(|g_t|)$). The results (Figure 11) quantitatively show that PCPO maintains more stable gradients.

We also added **Appendix E** to draw a direct theoretical analogy between our method and the credit assignment in REINFORCE.

**4. Reproducibility** We have released the anonymized core implementation code to facilitate verification of our method.

---

We have also expanded the preliminaries for accessibility, optimized the layout, refined experimental details to ensure reproducibility, analyzed wall-clock time, and added comparisons against SOTA DPO/Self-Play methods.

We believe these revisions significantly strengthen the paper and provide robust evidence that PCPO is a state-of-the-art, model-agnostic alignment method. We thank the reviewers again for helping us improve the quality of this work.

We have provided detailed responses to each reviewer below and look forward to any further discussion.

---

### Author Response · Authors · 2025-11-27
**Follow-up on Rebuttal and Revision**

Dear Reviewers,

We are writing to kindly check if you have had the opportunity to review our rebuttal and the updated paper. We have made extensive revisions to address your comments, which are highlighted in blue in the updated paper.

We value your time and feedback, and look forward to engaging in further discussion to resolve any remaining concerns.

Best regards,
Authors

---

### Author Response · Authors · 2025-12-01
**Summary of Discussion**

To the Newly Assigned Area Chair,

Thank you for handling our paper. To assist your review, we provide a summary of the discussion.

The **General Response** below summarizes (1) strengths pointed out by multiple reviewers, and (2) major weaknesses and questions which we have addressed. \
**Table 2 ($\text{FD}_\text{DINO}$)** (page 6), **Table 5** (page 8), **Figure 5** (page 9), **Table 9** (page 23), **Figure 9** (page 24), and **Figures 10, 11** (page 25) display new *quantitative* results and analyses. \
Other revisions strengthen our theoretical justification and/or provide more context.

All revisions that address reviewers' concerns are highlighted in *blue* in the updated manuscript, which was **uploaded on November 20** for sufficient discussion time. \
We report that **reviewer oEo3 had raised their score from 6 to 8** after reviewing our rebuttal and new results, providing evidence that our rebuttal was effective. Other reviewers had not yet responded before the system reset.

We believe our revisions have significantly strengthened the paper, providing robust evidence on the effectiveness of PCPO. We hope this summary aids your assessment.

Sincerely, \
Authors

---

### Meta-Review · Area_Chair_iYBP · 2026-01-05

**Summary:**

This paper works on RL to improve text-to-image generation models. Authors identifies a key cause of training instability and proposed PCPO to improve it. Experimental results verified the effectiveness of the proposed methods.

This paper got one 8, one 6 and two 4 ratings.

The strength of this paper given by reviewers are:
1. paper is well written. (Reviewer 4hys, gCoT)
2. paper is well motivated. (Reviewer 4hys)
3. good improvements. (Reviewer 4hys)
4. fast convergence and better quality compared with DanceGRPO. (Reviewer 71VF)
5. Novel problem formulation. (Reviewer oEo3)
6. Technically novelty. (Reviewer oEo3)
7. Strong empirical validation. (Reviewer oEo3, gCoT)

The weakness of this paper given by reviewers are:
1. better to have modern metrics for fidelity evaluation. (Reviewer 4hys, 71VF, oEo3)
2. provide some details. (Reviewer 4hys)
3. better to add introductory overview of diffusion models and GRPO. (Reviewer 71VF)
4. would like to see more complex qualitative comparisons in Appendix G. (Reviewer 71VF)
5. need to evaluated on widely used datasets. (Reviewer 71VF)
6. whether the reported conclusions remain consistent across different target levels. (Reviewer 71VF)
7. robustness and generalizability? (Reviewer 71VF, oEo3)
8. Theoretical Intuition. (Reviewer gCoT)
9. Lack of baselines. (Reviewer gCoT)

Questions:
1. add some details. (Reviewer 4hys)
2. Why does it cause a negative effect when over-weighting the higher noise region? (Reviewer 4hys)
3. Computation vs. Speedup. (Reviewer gCoT)
4. Variance Quantification. (Reviewer gCoT)
5. please show the distribution /rho_t? (Reviewer gCoT)

AC read authors' paper, reviewers' comment, authors' rebuttal and believe authors have addressed reviewers comments and thus decide to accept this paper.

**Reviewer Concerns:**

weakness 1. added results for FD-DINOv2

weakness 2. added more details.

weakness 3. added.

weakness 4. added more results.

weakness 5. added more results.

weakness 6. results have already been reported in the paper.

weakness 7. added more results.

weakness 8. added more explanation in Appendix E.

weakness 9. added more baseline.

question 1. added results.

question 2. added more explanation.

question 3. added in appendix D.

question 4. added in appendix D.

question 5. added in appendix C with new results.

**Reviewer Scores:**

Reviewer 4hys will keep score 8.

Reviewer 71VF might increase their score from 4.

Reviewer oEo3 might keep or increase their score 6.

Reviewer gCoT might increase their score from 4.

---

### Decision · Program_Chairs · 2026-01-26

Accept (Poster)